# Direct observation of the neural computations underlying a single decision

Natalie Steinemann[1†], Gabriel M Stine[1†§], Eric Trautmann[1], Ariel Zylberberg[1,2,3‡], Daniel M Wolpert[1,2,4‡], Michael N Shadlen[1,2,3,4*‡]

[1]Zuckerman Mind Brain and Behavior Institute, Columbia University, New York, United States; [2]Howard Hughes Medical Institute, Chevy Chase, United States; [3]Department of Neuroscience, Columbia University, New York, United States; [4]Kavli Institute, Columbia University, New York, United States

*For correspondence: shadlen@columbia.edu

[†]These authors contributed equally to this work
[‡]These authors also contributed equally to this work

Present address: [§]Department of Brain and Cognitive Science, MIT, Cambridge, United States

**Abstract** Neurobiological investigations of perceptual decision-making have furnished the first glimpse of a flexible cognitive process at the level of single neurons. Neurons in the parietal and prefrontal cortex are thought to represent the accumulation of noisy evidence, acquired over time, leading to a decision. Neural recordings averaged over many decisions have provided support for the deterministic rise in activity to a termination bound. Critically, it is the unobserved stochastic component that is thought to confer variability in both choice and decision time. Here, we elucidate this drift-diffusion signal on individual decisions. We recorded simultaneously from hundreds of neurons in the lateral intraparietal cortex of monkeys while they made decisions about the direction of random dot motion. We show that a single scalar quantity, derived from the weighted sum of the population activity, represents a combination of deterministic drift and stochastic diffusion. Moreover, we provide direct support for the hypothesis that this drift-diffusion signal approximates the quantity responsible for the variability in choice and reaction times. The population-derived signals rely on a small subset of neurons with response fields that overlap the choice targets. These neurons represent the integral of noisy evidence. Another subset of direction-selective neurons with response fields that overlap the motion stimulus appear to represent the integrand. This parsimonious architecture would escape detection by state-space analyses, absent a clear hypothesis.

## eLife assessment

This **fundamental** work quantifies the stochastic dynamics of neural population activity in the lateral intraparietal area (LIP) of the macaque monkey brain during single perceptual decisions. These single-trial dynamics have been subject to intense debate in neuroscience, and they have significant implications for modeling decision-making in various fields including neuroscience and psychology. Through a combination of state-of-the-art recordings from many LIP neurons and theory-driven data analyses, the authors provide **convincing** evidence for the notion that single-trial neural population dynamics in LIP encode the decision variable postulated by the drift-diffusion model of decision-making.

## Introduction

Neural signals in the mammalian cortex are notoriously noisy. They manifest as a sequence of action potentials (spikes) that approximate non-stationary Poisson point processes. Therefore, to characterize the signal produced by a neuron, electrophysiologists typically combine the spike times from many repetitions or trials relative to the time of an event (e.g., stimulus onset) to yield the average firing rate of the neuron as a function of time. Such trial-averaged firing rates are the main staple of

systems neuroscience (Here and throughout, trial average and across-trial average refer to the mean of signal values, over all specified trials at the same time, $t$, relative to a trial event (e.g., motion onset)). They are the source of knowledge about spatial selectivity (e.g., receptive fields), feature selectivity (e.g., direction of motion, faces vs. other objects), and even cognitive signals associated with working memory, anticipation, attention, motor planning, and decision-making. But there is an important limitation.

Trial averages suppress signals that vary independently across trials. In many cognitive tasks, such as difficult decisions, the variable component of the signal is the most interesting because it is this component that is thought to explain the variable choice and response time. This variability is thought to arise from a decision process that accumulates noisy evidence in favor of the alternatives and terminates when the accumulated evidence for one alternative, termed the decision variable (DV), reaches a terminating bound. The DV is stochastic because the integral of noisy samples of evidence is biased Brownian motion (or drift-diffusion) and this leads to a stochastic choice and response time on each decision. However, the stochastic part of this signal is suppressed by averaging across trials. We will use the term drift-diffusion because it is the expression most commonly applied in models of decision-making (*Ratcliff and Rouder, 1998*; *Gold and Shadlen, 2007*), and we will consider the noise part—that is, diffusion—as the signal of interest.

In the setting of difficult perceptual decisions, studied here, bounded drift-diffusion reconciles the relationship between decision speed and accuracy. It also explains the trial-averaged firing rates of neurons in the lateral intraparietal area (LIP) that represent the action used by monkeys to indicate their choice. These firing rate averages show motion-dependent, ramping activity that reflects the direction and strength of motion, consistent with the drift component of drift-diffusion (*Roitman and Shadlen, 2002*). Up to now, however, the diffusion component has not been observed, owing to averaging.

There is thus a missing link between the mathematical characterization of the decision process and its realization in neural circuits, leaving open the possibility that drift-diffusion dynamics do not underlie LIP activity (e.g., *Latimer et al., 2015*), or emerge only at the level of the population, without explicit representation by single neurons. We reasoned that these and other alternatives to drift-diffusion could be adjudicated if it were possible to resolve the DV giving rise to a single decision.

This stratagem is now feasible, owing to the development of high-density Neuropixels probes, which are capable of recording from deep sulci in the primate brain. Here we provide the first direct evidence for a drift-diffusion process underlying single decisions. We recorded simultaneously from up to 203 neurons in area LIP while monkeys made perceptual decisions about the direction of dynamic random dot motion (*Newsome et al., 1989*; *Gold and Shadlen, 2007*). Using a variety of dimensionality reduction techniques, we show that a drift-diffusion signal can be detected in such populations on individual trials. Moreover, this signal satisfies the criteria for a DV that controls the choice and reaction time (RT). Notably, the signal of interest is dominated by a small subpopulation of neurons with response fields that overlap one of the choice targets, consistent with earlier single-neuron studies (e.g., *Shadlen and Newsome, 1996*; *Roitman and Shadlen, 2002*; *Churchland et al., 2011*; *Gold and Shadlen, 2007*).

## Results

Two monkeys made perceptual decisions, reported by an eye movement, about the net direction of dynamic random dot motion (*Figure 1a*). We measured the speed and accuracy of these decisions as a function of motion strength (*Figure 1b*, circles). The choice probabilities and the distribution of RTs are well described (*Figure 1b*, traces) by a bounded drift-diffusion model (*Figure 1c*). On 50% of the trials, a brief (100 ms) pulse of weak leftward or rightward motion was presented at a random time. The influence of these pulses on choice and RT further supports the assertion that the choices and RTs arose through a process of integration of noisy samples of evidence to a stopping bound (*Figure 1— figure supplement 1*; *Stine et al., 2020*; *Stine et al., 2023*; *Hyafil et al., 2023*).

In addition to the main task, the monkeys performed two control tasks: instructed, delayed saccades to peripheral targets and passive viewing of random dot motion (see 'Methods'). These control tasks served to identify, post hoc, neurons with response fields that overlap the choice target in the hemifield contralateral to the recording site ($T_{in}^{con}$), neurons with response fields that overlap

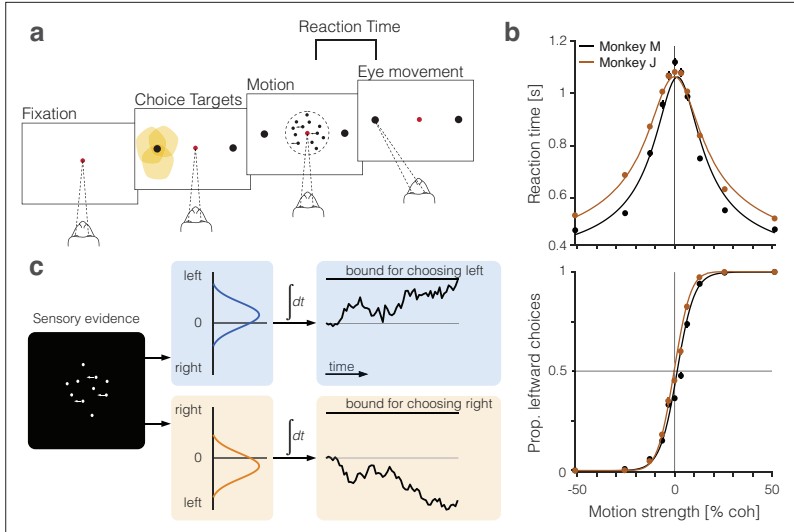

**Figure 1.** Perceptual decisions are explained by the accumulation of noisy evidence to a stopping bound. (**a**) Random dot motion discrimination task. The monkey fixates a central point. After a delay, two peripheral targets appear, followed by the random dot motion. When ready, the monkey reports the net direction of motion by making an eye movement to the corresponding target. Yellow shading indicates the response fields of a subset of neurons in lateral intraparietal cortex (LIP) that we refer to as $T_{in}$ neurons (*target in response field*). (**b**) Mean reaction times (top) and proportion of leftward choices (bottom) plotted as a function of motion strength and direction, indicated by the sign of the coherence: positive is leftward. Data (circles) are from all sessions from monkey M (black, 9684 trials) and monkey J (brown, 8142 trials). Solid lines are fits of a bounded drift-diffusion model. (**c**) Drift-diffusion model. The decision process is depicted as a race between two accumulators: one integrating momentary evidence for left; the other for right. The momentary samples of evidence are sequential samples from a pair negatively correlated Normal distributions with opposite means ($\rho = -0.71$). The decision is terminated when one accumulator reaches its positive bound. The example depicts leftward motion leading to a leftward decision.

The online version of this article includes the following figure supplement(s) for figure 1:

**Figure supplement 1.** Effect of motion pulses on behavior (adapted from Figure S1 of *Stine et al., 2023*).

---

the other choice target ($T_{in}^{ips}$), and neurons with response fields that overlap the random-dot motion stimulus ($M_{in}$; *Table 1*).

We recorded simultaneously from populations of neurons in area LIP using newly developed macaque Neuropixels probes (*Trautmann et al., 2023*) while monkeys performed these tasks. The data set comprises eight sessions from two monkeys (1696–2894 trials per session; *Table 2*). Our primary goal was to identify activity in LIP that relates to the DV, a theoretical quantity that determines the choice and RT on each trial. To achieve this, we formed weighted averages from all neurons in

**Table 1.** Information about individual experimental sessions.

| Session | 1 | 2 | 3 | 4 | 5 | 6 | 7 | 8 | Mean |
|---|---|---|---|---|---|---|---|---|---|
| Monkey | M | M | M | M | M | J | J | J | |
| Trials | 1797 | 1696 | 2256 | 1859 | 2076 | 2449 | 2799 | 2894 | 2228 |
| Neurons | 191 | 90 | 54 | 140 | 107 | 138 | 161 | 203 | 135.5 |
| %$T_{in}^{con}$ | 8.9 | 14.4 | 16.7 | 15 | 17.8 | 8.7 | 21.1 | 13.3 | 14.5 |
| %$T_{in}^{ips}$ | 4.2 | 5.6 | 13 | 13.6 | 6.5 | 12.3 | 0.6 | 2.0 | 7.2 |
| %$M_{in}^{left}$ | 5.2 | 5.6 | 0 | 5 | 3.7 | 2.9 | 3.7 | 7.4 | 4.2 |
| %$M_{in}^{right}$ | 1.0 | 2.2 | 3.7 | 1.4 | 2.8 | 3.6 | 2.5 | 3.0 | 2.5 |

**Table 2.** Model fit parameters.

$\kappa$: scaling of motion strength to drift rate; $B_0$: bound height; $\alpha$: linear urgency component; $\mu_{nd}$: mean of the non-decision time; $\sigma_{nd}$: standard deviation of the non-decision time; and $C_0$: bias.

| Parameter | $\kappa$ | $B_0$ | $\alpha$ | $\mu_{nd}$ | $\sigma_{nd}$ | $C_0$ |
|---|---|---|---|---|---|---|
| Monkey M | 13.37 | 1.03 | 0.4199 | 0.317 | 0.039 | –0.0144 |
| Monkey J | 13.72 | 1.76 | 1.3591 | 0.291 | 0.055 | 0.0008 |

the sample population, including those with response fields that do not overlap a choice target or the motion stimulus. We used several strategies to assign this vector of weights, which we refer to as a *coding direction* in the neuronal state space (NSS). The projection of the spiking activity from the population of neurons onto the vector of weights gives rise to a scalar function of time, $S^x(t)$, where the superscript $x$ labels the strategy. We focus on such one-dimensional projections because of the long-standing hypothesis that the DV is drift-diffusion, which is a scalar function of time.

We first developed a targeted strategy that would reproduce the well-known coherence-dependent ramping activity evident in the across-trial averages. This strategy applies regression to best approximate a linear ramp, on each trial, $i$, that terminates with a saccade to the choice target contralateral to the hemisphere of the LIP recordings. The ramps are defined on the epoch spanning the decision time: from $t_0 = 0.2$ s after motion onset to $t_1 = 0.05$ s before saccade initiation (black lines in *Figure 2— figure supplement 1*) The epoch is motivated by many previous studies (see *Gold and Shadlen, 2007*; *Shadlen and Kiani, 2013*, for reviews). Each ramp begins at $f_i(t_0) = -1$ and ends at $f_i(t_1) = 1$. The ramp approximates the expectation—conditional on the choice and response time—of the deterministic components of the drift-diffusion signal, which, in addition to the drift, can incorporate (i) a time-dependent but evidence-independent urgency signal (*Churchland et al., 2008*; *Drugowitsch et al., 2012*), and (ii) a dynamic bias signal (*Hanks et al., 2011*). It can also be viewed as an approximation to firing rates averaged across trials and grouped by contraversive choice and RT quantile (*Figure 2—figure supplement 2*). Importantly, the fit is not guided by an assumption of an underlying diffusion process. That is, the ramp coding direction is agnostic to the underlying processes whose averages approximate ramps. The weights derived from these regression fits specify a *ramp coding direction* in the state space defined by the population of neurons in the session. The single-trial signal, $S^{ramp}(t)$, is rendered as the projection of the population firing rates onto this coding direction.

The left side of *Figure 2a* shows single-trial activity rendered by this strategy. The right side of the figure shows the averages of the single-trial responses grouped by signed coherence and aligned to motion onset or response time (saccade initiation). These averaged traces exhibit features of the firing rate averages in previous studies of single neurons in LIP (e.g., *Roitman and Shadlen, 2002*). They begin to diverge as a function of the direction and strength of motion approximately 200 ms after the onset of motion. The traces converge near the time of saccadic response to the contralateral choice target such that the coherence dependence is absent or greatly diminished. Coherence dependence remains evident through the initiation of saccades to the right (ipsilateral) target, consistent with a race architecture—between negatively correlated accumulators—depicted in *Figure 1c*.

We complemented this regression strategy with principal component analysis (PCA) and use the first PC (PC1), which explains $44 \pm 7\%$ of the variance (mean ± s.e. across sessions) of the activity between 200 and 600 ms from motion onset (see 'Methods'). This coding direction renders single-trial signals, $S^{PC1}(t)$ (*Figure 2b*). In a third strategy, we consider the mean activity of neurons with response fields that overlapped the contralateral choice target ($T_{in}^{con}$ neurons), which were the focus of previous single-neuron studies (e.g., *Shadlen and Newsome, 1996*; *Platt and Glimcher, 1999*; *Roitman and Shadlen, 2002*). In those studies, the task was modified so that one of the choice targets was placed in the neural response field, whereas here we identify neurons post hoc with response fields that happen to overlap the contralateral choice target. This difference probably accounts for the lower firing rates of the $T_{in}^{con}$ neurons studied here. *Figure 2c* shows single-trial and across-trial averages from these $T_{in}^{con}$ neurons. They too render signals, $S_{Tin}^{con}(t)$, similar to those derived from the full population. The $T_{in}^{con}$ neurons thus furnish a third coding direction defined by a vector of identical positive weights assigned to all $T_{in}^{con}$ neurons and 0 for all other neurons in the population. The emboldened single-trial traces in *Figure 2* (left) correspond to the same trials rendered by the three coding directions. It is not difficult

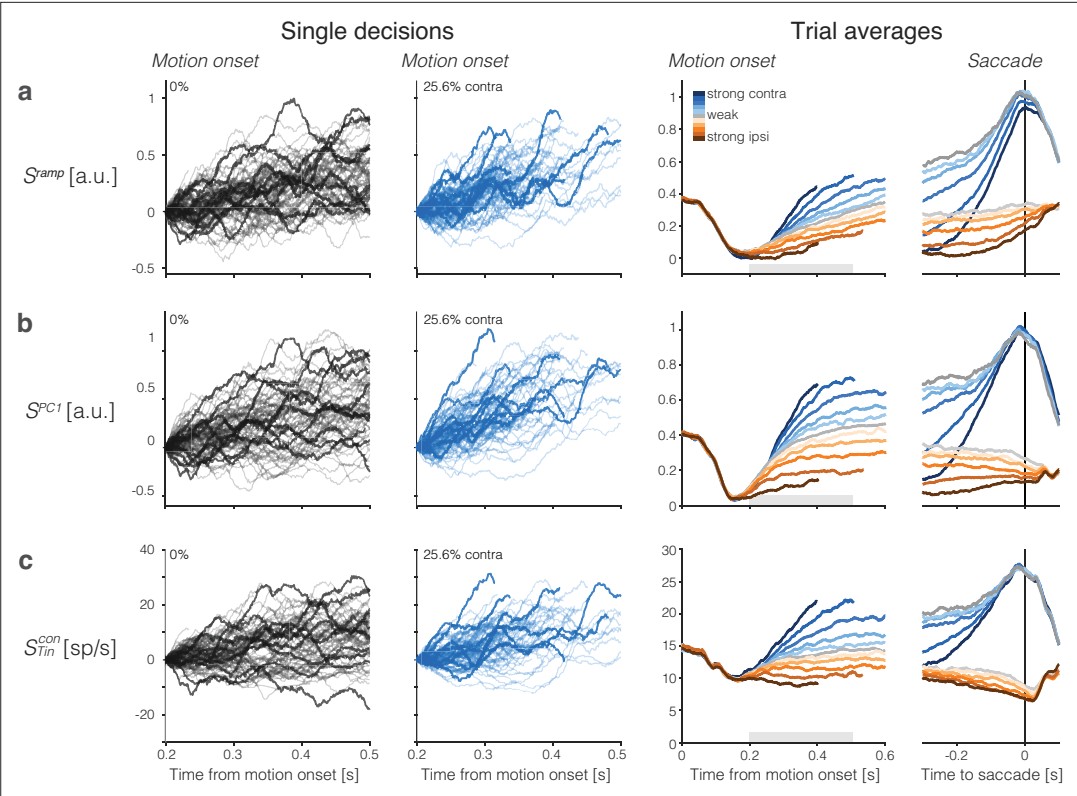

**Figure 2.** Population responses from lateral intraparietal cortex (LIP) approximate drift-diffusion. Rows show three types of population signals. The left columns show representative single-trial firing rates during the first 300 ms of evidence accumulation using two motion strengths: 0 and 25.6% coherence toward the left (contralateral) choice target. For visualization, single-trial traces were baseline corrected by subtracting the activity in a 50 ms window around 200 ms. We highlight several trials with thick traces (same trials in **a–c**). The right columns show the across-trial average responses for each coherence and direction. Motion strength and direction are indicated by color (legend) and aligned to motion onset (left) or saccade initiation (right). The gray bars under the motion-aligned averages indicate the 300 ms epoch used in the display of the single-trial responses (left panels). The epoch begins when LIP first registers a signal related to the strength and direction of motion. Except for saccade-aligned response, trials are cut off 100 ms before saccade initiation. Error trials are excluded from the saccade-aligned averages, only. (**a**) Ramp coding direction. The weight vector is established by regression to ramps from -1 to +1 over the period of putative integration, from 200 ms after motion onset to 100 ms before saccade initiation (see *Figure 2—figure supplement 1*). Only trials ending in left (contralateral) choices are used in the regression. (**b**) First principal component (PC1) coding direction. (**c**) Average firing rates of the subset of neurons that represent the left (contralateral) target. The weight vector consists of $\frac{1}{N}$ for each of the $N$ $\mathrm{T_{in}^{con}}$ neurons and 0 for all other neurons. Note the similarity of both the single-trial traces and the response averages produced by the different weighting strategies.

The online version of this article includes the following figure supplement(s) for figure 2:

**Figure supplement 1.** Derivation of a ramp coding direction in neuronal state space.

**Figure supplement 2.** Trial-averaged activity grouped by reaction time (RT) quantile.

**Figure supplement 3.** Trial-averaged activity after subtracting the urgency component.

---

to tell which are the corresponding traces, an observation that speaks to their similarity, and the same is true for the averages. We will expand on this observation in what follows.

The averages show the deterministic *drift* component of the hypothesized drift-diffusion process, with the slope varying monotonically with the signed motion strength (*Figure 2*, right). The rise begins to saturate as a consequence of the putative termination bound—a combination of dropout of trials that are about to terminate and the effect on the distribution of possible diffusion paths imposed by the very existence of a stopping bound. This saturation is evident earlier on trials with stronger motion, hence shorter RT, on average. The positive buildup rate on the 0% coherence motion represents

the time-dependent, evidence-independent signal that is thought to reflect the cost of time. It leads to termination even if the evidence is weak, equivalent to collapsing stopping bounds in traditional, symmetric drift-diffusion models (*Drugowitsch et al., 2012*). Removal of this *urgency* signal, $u(t)$, from the non-zero coherence traces renders the positive and negative coherence averages symmetric relative to zero on the ordinate (*Figure 2—figure supplement 3*).

The single-trial responses in *Figure 2* do not look like the averages but instead approximate drift-diffusion. We focus on the epoch from 200 to 500 (or 600) ms from motion onset—that is, the first 300 (or 400) ms of the period in which the averages reflect the integration of evidence. Some traces are cut off before the end of the epoch because a saccade occurred 100 ms later on the trial. However, most 0% coherence trials continue beyond 500 ms (median RT > 600 ms). The single-trial traces do not rise monotonically as a function of time but meander and tend to spread apart from each other vertically. For unbounded diffusion, the variance would increase linearly, but as just mentioned, the existence of an upper stopping bound and the limited range of firing rates (e.g., non-negative) renders the function sublinear at later times (*Figure 3a*, *Figure 3—figure supplement 1*). The autocorrelation between an early and a later sample from the same diffusion trace is also clearly specified for unbounded diffusion. The theoretical values shown in *Figure 3b and c* are the autocorrelations of unbounded diffusion processes that are smoothed identically to the neural signals (see 'Methods' and Appendix 1). The autocorrelations in the data follow a strikingly similar pattern. These observations support the assertion that the coherence-dependent (ramp-like) firing rate averages observed in previous studies of area LIP are composed of stochastic drift-diffusion processes on single trials.

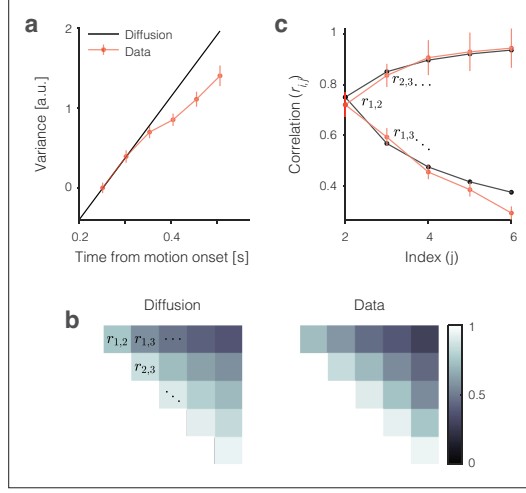

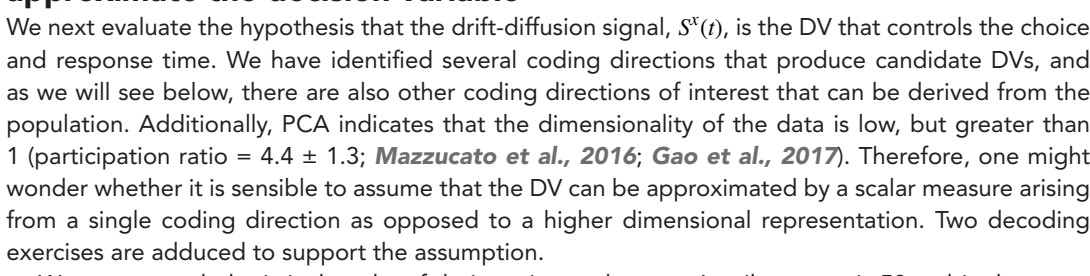

**Figure 3.** Variance and autocorrelation of the single-trial signals. The analyses here are based on samples of $S^{\mathrm{ramp}}$ at six time points during the first 300 ms of putative integration, using all 0% and ±3.2% coherence trials ($N = 5927$). Samples are separated by the width of the boxcar filter (51 ms), beginning at $t_1 = 226$ ms. (**a**) Variance increases as a function of time. The measure of variance is normalized so that it is 1 for the first sample. Error bars are s.e. (bootstrap). (**b**) Autocorrelation of samples as a function of time and lag approximate the values expected from diffusion. The upper triangular portion of the 6 × 6 correlation matrix for unbounded diffusion ($r_{i,j}$ is represented by brightness). The values from the data ($S^{\mathrm{ramp}}$) are similar (right). (**c**) Nine of the 15 autocorrelation terms in (**b**) permit a more direct comparison of theory and data. The lower limb of the C-shaped function shows the decay in $r_{i,j}$ as a function of lag ($j - i$). This is the top row of (**b**). The upper limb shows the increase in $r_{i,j}$ as a function of time (for fixed lag). This is the lower diagonal in (**b**). Error bars are s.e. (bootstrap). Note that the autocorrelations incorporate a free parameter, $\phi \le 1$, that serves to correct for an unknown fraction of the measured variance that is not explained by diffusion (see 'Methods').

The online version of this article includes the following figure supplement(s) for figure 3:

**Figure supplement 1.** Bounds induce sublinear increase in variance of diffusion paths.

## Single-trial drift-diffusion signals approximate the decision variable

We next evaluate the hypothesis that the drift-diffusion signal, $S^x(t)$, is the DV that controls the choice and response time. We have identified several coding directions that produce candidate DVs, and as we will see below, there are also other coding directions of interest that can be derived from the population. Additionally, PCA indicates that the dimensionality of the data is low, but greater than 1 (participation ratio = 4.4 ± 1.3; *Mazzucato et al., 2016*; *Gao et al., 2017*). Therefore, one might wonder whether it is sensible to assume that the DV can be approximated by a scalar measure arising from a single coding direction as opposed to a higher dimensional representation. Two decoding exercises are adduced to support the assumption.

We constructed a logistic decoder of choice using each neuron's spike counts in 50 ms bins between 100 and 500 ms after motion onset. As shown in *Figure 4a*, this *What*-decoder (orange) predicts

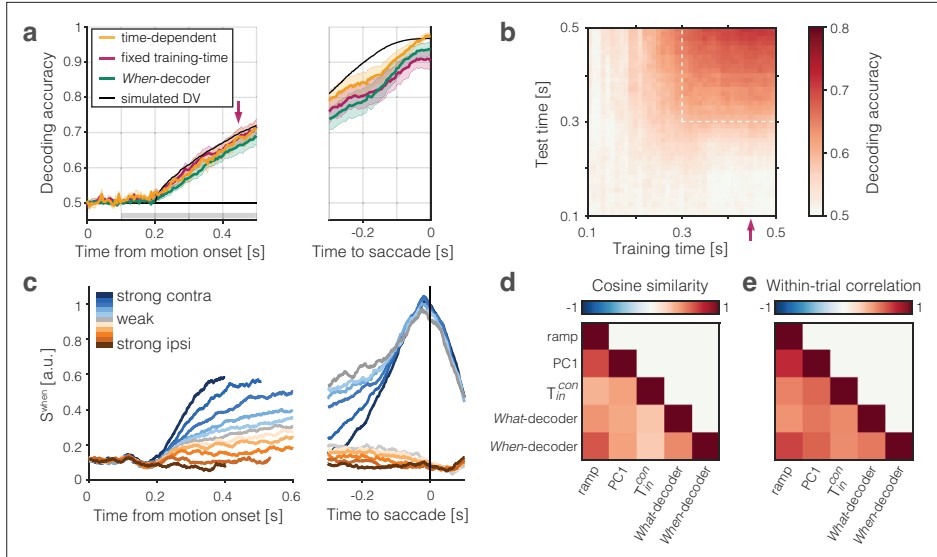

**Figure 4.** The population signal predictive of choice and reaction time (RT) is approximately one-dimensional. Two binary decoders were trained to predict the choice (*What*-decoder) and its time (*When*-decoder) using the population responses in each session. The *When*-decoder predicts whether a saccadic response to the contralateral target will occur in the next 150 ms, but critically, its accuracy is evaluated based on its ability to predict choice. (**a**) Cross-validated choice decoding accuracy plotted as a function of time from motion onset (left) and time to saccadic choice (right). Values are averages across sessions. The *What*-decoder is either trained at the time point at which it is evaluated (time-dependent decoder, orange) or at the single time point indicated by the red arrow ($t = 450$ ms after motion onset; *fixed training-time* decoder, red). Both training procedures achieve high levels of accuracy. The *When*-decoder is trained to capture the time of response only on trials terminating with a left (contraversive) choice. The coding direction identified by this approach nonetheless predicts choice (green) nearly as well as the *fixed training-time What*-decoder. The black trace shows the accuracy of a *What*-decoder trained on simulated signals using a drift-diffusion model that approximates the behavioral data in *Figure 1*. Error bars signify s.e.m. across sessions. The gray bar shows the epoch depicted in the next panel. (**b**) The heat map shows the accuracy of a decoder trained at times along the abscissa and tested at times along the ordinate. Time is relative to motion onset (gray shading in **a**). In addition to training at $t = 450$ ms, the decoder can be trained at any time from $300 < t < 500$ ms (dashed box) and achieve the same level of accuracy when tested at any single test time. The orange and red traces in **a** correspond to the main diagonal ($x = y$) and the column marked by the red arrow, respectively. (**c**) Trial-averaged activity rendered by the projection of the population responses along the *When* coding direction, $S^{\text{When}}$. Same conventions as in *Figure 2*. (**d**) Cosine similarity of five coding directions. The heatmap shows the mean values across the sessions, arranged like the lower triangular portion of a correlation matrix. Cosine similarities are significantly greater than zero for all comparisons (all p<0.001, *t*-test). (**e**) Correlation of single-trial diffusion traces. The Pearson correlations are calculated from ordered pairs of $\{S_i^x(t), S_i^y(t)\}$, where $S_i(t)$ are the detrended signals rendered by coding directions, $x$ and $y$, on trial $i$. The detrending removes trial-averaged means for each signed coherence, leaving only the diffusion component. Reported correlations are significantly greater than zero for all pairs of coding directions and sessions (all p<10$^{-23}$, *t*-test, see 'Methods'). The variability in cosine similarity and within-trial correlation across sessions is portrayed in *Figure 4—figure supplement 3*.

The online version of this article includes the following figure supplement(s) for figure 4:

**Figure supplement 1.** Weights are assigned to each of the $N$ simultaneously recorded neurons in each session using logistic regression to approximate a step that takes a value of 0 from 200 ms before motion onset to 150 ms before saccade initiation, and a value of 1 for the following 100 ms (the last 50 ms before the saccade are discarded).

**Figure supplement 2.** Single-trials and trial-averaged signals furnished by the When- and What-decoders.

**Figure supplement 3.** Variability in cosine similarity and within-trial correlation across sessions.

**Figure supplement 4.** Comparison of linear and non-linear choice decoders.

**Figure supplement 5.** Decoding choice from subsets of neurons.

choice as accurately as a decoder of simulated data from a drift-diffusion model (black) using parameters derived from fits to the monkeys' choice and RT data (see 'Methods'). The simulation establishes a rough estimate of the decoding accuracy that can be achieved, given the stochastic nature of the choice, were we granted access to the drift-diffusion signal that actually determines the decision. In this analysis, the decoder can use a different vector of weights at each point in time (*time-dependent* coding directions; see *Peixoto et al., 2021*). However, if the representation of the DV in LIP is one-dimensional, then a decoder trained at one time should perform well when tested at a different time. The red curve in *Figure 4a* shows the performance of a *What*-decoder with a *fixed training-time* (450 ms after motion onset; red arrow). This decoder performs nearly as well as the decoder trained at each time bin. The heatmap (*Figure 4b*) generalizes this observation. It shows two main features for all times $300 < t < 500$ ms (dashed box). First, unsurprisingly, for a *What* choice decoder trained on data at one time $t = x$, the predictions improve as the testing time advances (the decoding accuracy increases along any vertical) as more evidence is accrued. Second, and more importantly, decoders tested at time $t = y$ perform similarly, independent of when they were trained (there is little variation in decoding accuracy along any horizontal). This observation suggests that a single vector of weights may suffice to decode the choice from the population response.

The second decoder is trained to predict whether a saccade to the contralateral choice target will be initiated in the next 150 ms. This *When*-decoder is trained by logistic regression to predict a binary output: 1 at all time-points that are within 150 ms of an upcoming saccade and 0 elsewhere (*Figure 4—figure supplement 1*). We validated the When-decoder by computing the area under an ROC (AUC) using the held-out (odd) trials (mean AUC over all time points: 0.84), but this is tangential to our goal. Although the *When*-decoder was trained only to predict the time of saccades, our rationale for developing this decoder was to test whether the *When* coding direction can be used to predict the *choice*. The green trace in *Figure 4a* shows the accuracy of $S^{\text{When}}(t)$ to predict the choice. The performance is almost identical to the choice decoder, despite being trained on a temporal feature of trials ending in the same left choice. This feat is explained by the similarity of signals produced by the *When*- and other coding directions. Note the similarity of the trial-averaged $S^{\text{When}}$ signals displayed in *Figure 4c* to those in *Figure 2* (see also *Figure 4—figure supplement 2*, right). Indeed, the cosine similarity between the *When* and *Ramp* coding directions is $0.67 \pm 0.03$ (*Figure 4d*). In light of this, it is not surprising that the weighting vectors derived from both the *What*- and *When*-decoders also render single-trial drift-diffusion traces that resemble each other and those rendered by other coding directions (*Figure 4e*). Together these analyses support the assertion that the DV is likely to be captured by a single dimension, consistent with *Ganguli et al., 2008*.

If the one-dimensional signals, $S^x(t)$, approximate the DV, they should explain the variability of choice and RT for trials sharing the same direction and motion strength. Specifically, (i) early samples of $S^x(t)$ should be predictive of choice and correlate inversely with the RT on trials that result in contraversive (leftward) choices, (ii) later samples ought to predict choice better and correlate more strongly (negatively) with RT than earlier samples, and (iii) later samples should contain the information present in the earlier samples and thus mediate (i.e., reduce the leverage) of the earlier samples on choice and RT. Each of these predictions is borne out by the data.

The analyses depicted in *Figure 5* allow us to visualize the influence of the single-trial signals, $S^x(t)$, on the choice and RT on that trial. We focus on the early epoch of evidence accumulation (200–550 ms after random dot motion onset) and restrict the analyses to decisions with $\text{RT} \geq 670$ ms and $\text{coherence} \leq 6.4\%$. The RT restriction eliminates 17% of the eligible trials. Larger values of $S^x(t)$ are associated with a larger probability of a left (contraversive) choice and a shorter RT, hence negative correlation between $S^x(t)$ and RT. We use the term, leverage, to describe the strength of both of these associations. The leverage on choice (*Figure 5a*, black traces) is the contribution of $S^x(t)$ to the log odds of a left choice, after accounting for the motion strength and direction (i.e., the coefficients, $\beta_1(t)$ in *Equation 8*). The leverage on RT (*Figure 5b*) is the Pearson correlation between $S^x(t)$ and the RT on that trial, after accounting for the effect of motion strength and direction on $S^x$ and RT (see 'Methods'). The leverage is evident from the earliest sign of evidence accumulation, 200 ms after motion onset, and its magnitude increases as a function of time, as evidence accrues (*Figure 5*, top). The filled circle to the right of the traces in each graph shows the leverage of $S^x$ at $t = 550$ ms, which is 120 ms before any of the included trials have terminated. Both observations are consistent with the hypothesis that $S^x$ represents the integral of noisy evidence used to form and terminate the

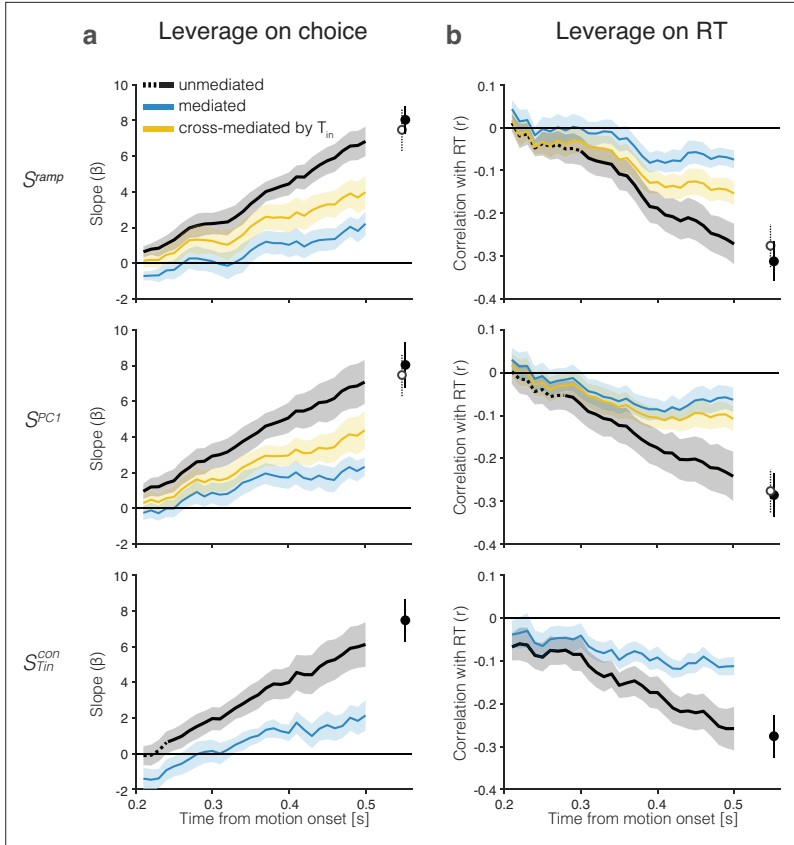

**Figure 5.** The drift-diffusion signal approximates the decision variable. The graphs show the leverage of single-trial drift-diffusion signals on choice and reaction time (RT) using only trials with $RT \geq 0.67$ s. Rows correspond to the same coding directions as in **Figure 2**. The graphs also demonstrate a reduction of the leverage of the samples at $t \leq 0.5$ s by a later sample of the signal at $t = 0.55$ s. Error bars are s.e.m. across sessions. (**a**) Leverage of single-trial drift-diffusion signals on choice. Leverage is the value of $\beta_1$, the coefficient that multiplies $S^x(t)$ in **Equation 8**. The black traces show the increase in leverage as a function of time. The dashed linestyle at the left end of three of the traces indicate values that are not statistically significant (p>0.05, bootstrap shuffle test, see 'Methods'). Filled symbols show the leverage at $t = 0.55$ s. The blue curve (*mediated*) shows the leverage when the later sample is included in the regression (**Equation 9**). Open symbols show the leverage of $S^{con}_{Tin}$ at $t = 0.55$ s (same value as the filled symbol in *bottom row*). The yellow curves (top and middle rows) show the leverage of the *cross-mediated* signal by $S^{con}_{Tin}(0.55)$. For all three signals, $S^x(t)$, leverage at $t = 0.4$ s is significantly mediated by $S^x(0.55)$ and cross-mediated by $S^{con}_{Tin}(0.55)$ (all p<$10^{-5}$, paired samples *t*-test). (**b**) Leverage of single-trial drift-diffusion signals on response time. Same conventions as in (**a**). Leverage is the correlation between $S^x(t)$ and RT. The mediated leverage is the partial correlation, given the later sample. For all three signals, $S^x(t)$, leverage at $t = 0.4$ s is significantly mediated by $S^x(0.55)$ and cross-mediated by $S^{con}_{Tin}(0.55)$ (all p<$10^{-8}$, paired samples *t*-test).

The online version of this article includes the following figure supplement(s) for figure 5:

**Figure supplement 1.** Control analyses bearing on the leverage and mediation results in **Figure 5**.

**Figure supplement 2.** The $T_{in}$ neurons are not discoverable by their weight assignments.

**Figure supplement 3.** Cross-mediation of single-trial correlations with behavior.

decision. Two control analyses demonstrate that the degree of leverage on choice and RT do not arise by chance: (i) random coding directions in state space produce negligible leverage (**Figure 5—figure supplement 1**, top), and (ii) breaking the trial-by-trial correspondence between neural activity and behavior eliminates all leverage (see Reviewer Figure 1 in reply to peer review).

Importantly, the leverage at earlier times is mediated by the later sample at $t = 550$ ms. The blue traces in all graphs show the remaining leverage once this later sample is allowed to explain the choice and RT—by including, respectively, an additional term in the logistic regression (**Equation 9**) and calculating the partial correlation, conditional on $S^x(t = 0.55)$. We assessed statistical significance

of the mediation statistics, $\xi^{\text{Ch}}$ and $\xi^{\text{RT}}$ (*Equations 7 and 10*) in each session for the three signals shown in *Figure 5* using a bootstrap procedure (see 'Methods', *Equation 10*). Mediation is significant in 47 of the 48 comparisons (all p<0.023, median p<$10^{-317}$). The one non-significant comparison is $\xi^{\text{Ch}}$ for $S_{\text{Tin}}^{\text{con}}$ in session 2 (p=0.73). The mediation is significant when this comparison is included in the combined data (p<$10^{-9}$, paired samples *t*-test). The stark decrease in leverage is consistent with one-dimensional diffusion in which later values of the signal contain the information in the earlier samples plus what has accrued in the interim. Had we recorded from all the neurons that represent the DV, we would expect the mediation to be complete (e.g., partial correlation = 0). However, our recorded population is only a fraction of the entire population. Indeed, the observed degree of mediation is similar to values obtained from simulations of weakly correlated, noisy neurons (*Figure 5—figure supplement 1*, bottom).

There is one additional noteworthy observation in *Figure 5* that highlights the importance of the $\text{T}_{\text{in}}^{\text{con}}$ neurons. The top and middle rows ($S^{\text{ramp}}$ and $S^{\text{PC1}}$) contain a second, open symbol, which is simply a copy of the filled symbol from the bottom row ($S_{\text{Tin}}^{\text{con}}$). The yellow traces show significant cross-mediation of $S^{\text{ramp}}$ and $S^{\text{PC1}}$ by the sample, $S_{\text{Tin}}^{\text{con}}(t = 0.55)$ (all p<0.05; median p<$10^{-268}$; bootstrap as above). This signal, carried by 9–21% of the neurons, mediates signals produced by the full population of 54–203 neurons nearly as strongly as $S^{\text{ramp}}$ and $S^{\text{PC1}}$ mediate themselves. The observation suggests that minimal leverage is gained by sophisticated analyses of the full NSS compared to a simple average of $\text{T}_{\text{in}}^{\text{con}}$ neurons. This is both reassuring and disquieting: reassuring because the $\text{T}_{\text{in}}^{\text{con}}$ neurons compose the dominant projection from LIP to the portions of the superior colliculus (SC) and the frontal eye field involved in the generation of saccades toward the contralateral choice target (*Paré and Wurtz, 1997*; *Ferraina et al., 2002*); disquieting because the functional relevance of these neurons is not revealed by the other coding directions. The weights assigned to the $\text{T}_{\text{in}}^{\text{con}}$ neurons span all percentiles (mean IQR: 49–96; mean 71st percentile, $AUC = 0.74 \pm 0.05$) in the ramp coding direction. They contribute disproportionately to PC1 and the *What-* and *When-* decoders but not enough to stand out based on their weights. Indeed, the ability to predict that a neuron is $\text{T}_{\text{in}}^{\text{con}}$ from its weight or percentile is remarkably poor (*Figure 5—figure supplement 2*).

These observations support the idea that the single-trial signals, $S^{\text{ramp}}$, $S^{\text{PC1}}$, and $S_{\text{Tin}}^{\text{con}}$, approximate the DV used by the monkey to make its decision. In *Figure 5—figure supplement 3*, we show that the $S^{\text{What}}$ and $S^{\text{When}}$ coding directions achieve qualitatively similar results. Moreover, a late sample from $S^x(t)$ mediates the earlier correlation with RT and choice of signals rendered by other coding directions, $S^y(t)$, at earlier times. Such cross-mediation is consistent with the high degree of cosine similarity between the coding directions (*Figure 4d*). The observation suggests that the DV is a prominent signal in LIP, discoverable by a variety of strategies, and consistent with the idea that it is one-dimensional. In *Figure 4—figure supplement 4*, we show that linear and nonlinear decoders achieve similar performance, which argues against a non-linear embedding of the DV in the population activity.

## Activity of direction-selective neurons in area LIP resembles momentary evidence

Up to now, we have focused our analyses on resolving the DV on single trials, paying little attention to how it is computed or to other signals that may be present in the LIP population. The drift-diffusion signal approximates the accumulation, or integral, of the noisy momentary evidence—a signal approximating the difference in the firing rates of direction-selective (DS) neurons with opposing direction preferences (e.g, in area MT; *Britten et al., 1996*). DS neurons, with properties similar to neurons in MT, have also been identified in area LIP (*Freedman and Assad, 2006*; *Shushruth et al., 2018*; *Fanini and Assad, 2009*; *Bollimunta and Ditterich, 2012*), where they are proposed to play a role in motion categorization (*Freedman and Assad, 2011*). We hypothesize that such neurons might participate in routing information from DS neurons in MT/MST to those in LIP that contain a choice target in their response fields.

We identified such DS neurons using a passive motion viewing task (*Figure 6a and b*, left). Neurons preferring leftward or rightward motion constitute 5–10% of the neurons in our sample populations *Table 1*. *Figure 6* shows the average firing rates of 51 leftward-preferring neurons ($\text{M}_{\text{in}}^{\text{left}}$, *Figure 6a*) and 26 rightward-preferring neurons ($\text{M}_{\text{in}}^{\text{right}}$, *Figure 6b*) under passive motion viewing and decision-making. The separation of the two traces in the passive viewing task is guaranteed because we used this task to identify the DS neurons. It is notable, however, that the DS is first evident about 100 ms

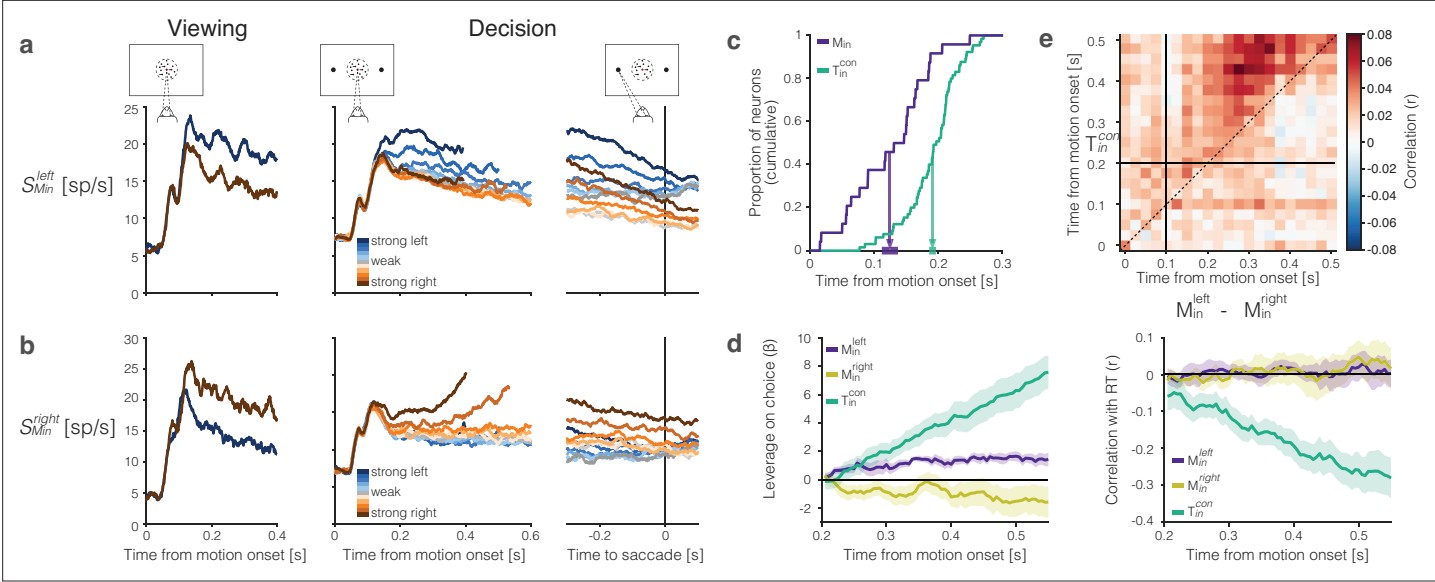

**Figure 6.** The representation of momentary evidence in area lateral intraparietal cortex (LIP). (**a**) Leftward preferring neurons. *Left*, response to strong leftward (blue) and rightward (brown) motion during passive viewing. Traces are averages over neurons and trials. The neurons were selected for analysis based on this task, hence the stronger response to leftward is guaranteed. Note the short-latency visual response to motion onset followed by the direction-selective (DS) response beginning ~100 ms after motion onset. *Right*, Responses during decision-making, aligned to motion onset and the saccadic response. Response averages are grouped by direction and strength of motion (color legend). The neurons retain the same direction preference during passive viewing and decision-making. The responses are also graded as a function of motion strength. (**b**) Rightward preferring neurons. Same conventions as (**a**). (**c**) Cumulative distribution of the times at which individual neurons start showing evidence-dependent activity. Evidence dependence emerges earlier in $M_{in}^{left}$ and $M_{in}^{right}$ neurons (purple) than in $T_{in}^{con}$ neurons (green). Arrows indicate the mean onset of evidence-dependent activity in each signal. The markers at the end of the arrows show the s.e.m. across neurons. (**d**) *Left*, leverage of neural activity on choice for $T_{in}^{con}$ (green), $M_{in}^{left}$ (purple), and $M_{in}^{right}$ (yellow) neurons. *Right*, same as *left*, for the correlation between neural activity and reaction time. The absence of negative correlation is explained by insufficient power (see 'Methods'). (**e**) Correlation between the neural representation of motion evidence—the difference in activity of neurons selective for leftward and rightward motion $\left(M_{in}^{left} - M_{in}^{right}\right)$—and the neural representation of the decision variable $\left(S_{Tin}^{con}\right)$ across different time points and lags. Horizontal and vertical lines indicate the onset of evidence-dependent activity in each signal. Positive correlations in the upper-left triangle indicate that the decision variable at a time point is correlated with earlier activity of the evidence signal.

after the onset of random dot motion, and this latency is also apparent in mean firing rates grouped by signed coherence during decision making (*Figure 6a and b* right). The activity of DS neurons is modulated by both the direction and strength of motion. However, unlike the $T_{in}$ neurons, the traces associated with different motion strengths are mostly parallel to one another and do not reach a common level of activity before the saccadic eye movement (i.e., they do not signal decision termination).

In addition to their shorter onset latency, the direction-selectivity of $M_{in}$ neurons precedes the choice-selectivity of $T_{in}^{con}$ neurons by ~100 ms (*Figure 6c*). The responses bear similarity to DS neurons in area MT. Such neurons are known to exhibit choice-dependent activity insofar as they furnish the noisy evidence that is integrated to form the decision (*Britten et al., 1996*; *Shadlen et al., 1996*). We computed putative single trial direction signals by averaging the responses from the left- and right-preferring DS neurons, respectively. The resulting signals, $S_{Min}^{left}(t)$ and $S_{Min}^{right}(t)$, have weak leverage on choice, but the leverage does not increase as a function of time (*Figure 6d*, left). This is what would be expected if the $M_{in}$ neurons represent the noisy momentary evidence as opposed to the accumulation thereof (*Mazurek et al., 2003*). We failed to detect a correlation between RT and either $S_{Min}$ signal (*Figure 6d*, right). This is surprising, but it could be explained by lack of power—a combination of small numbers of $M_{in}$ neurons, narrow sample windows (50 ms boxcar) and the focus on the long RT trials. Indeed, we found a weak but statistically significant negative correlation between RT and the difference in leftward vs. rightward signals, averaged over the epoch $0.1 \leq t \leq 0.4\,s$ from motion onset ($p=0.0004$; $\mathcal{H}_0 : \rho \geq 0$, see 'Methods').

We considered the hypothesis that these DS signals are integrated by the $T_{in}^{con}$ neurons to form the DV. The heatmap in *Figure 6e* supports this hypothesis. On each trial, we formed the ordered pairs, $\{x, y\}$, where $x = \tilde{S}_{Min}^{left}(t_x) - \tilde{S}_{Min}^{right}(t_x)$ and $y = \tilde{S}_{Tin}^{con}(t_y)$. The tilde in these expressions indicates the use of

standardized residual values, for each signed motion strength. The heatmap shows the correlation of these quantities across trials. If the hypothesis were true, the correlations should be positive for $t_y > t_x$ when $t_x > 100$ ms and $t_y > 200$ ms, and if the operation approximates integration, the level of correlation should be consistent at all lags, $t_y - t_x > 100$ ms. The correlations are significant in the epoch of interest, and they differ significantly from the average correlations in the rest of the graph (i.e., $t_x < 100$, $t_y < 200$, or $t_y < t_x$, p<0.0001 permutation test). Although correlative, the observation is consistent with the idea that evidence integration occurs within area LIP, rather than inherited from another brain area (*Zhang et al., 2022*; *Bollimunta and Ditterich, 2012*).

## Discussion

We have observed a neural representation of the stochastic process that gives rise to a single decision. This is the elusive drift-diffusion signal that has long been thought to determine the variable choice and response time in the perceptual task studied here. The signal was elusive because it is the integral of noisy momentary evidence, hence stochastic, and undetectable in the firing rates when they are computed as averages over trials. The averages preserve the ramp-like *drift* component, leaving open the possibility that the averages are composed of other stochastic processes (e.g., *Latimer et al., 2015*; *Cisek et al., 2009*). By providing access to populations of neurons in LIP, macaque Neuropixels probes (*Trautmann et al., 2023*) allowed us to resolve, for the first time, the evolution of LIP activity during a single decision.

The present findings establish that the ramp-like averages arise from drift-diffusion on single trials, and this drift-diffusion signal approximates the DV that arbitrates the choice and RT on that trial. We used a variety of strategies to assign a weight to each neuron in the population such that the vector of weights defines a coding direction in NSS. The weighted averages render population firing rate signals on single trials. Our experience is that any method of assigning the weights that captures a known feature of evidence accumulation (or its termination in a saccadic choice) reveals drift-diffusion on single trials, and this also holds for data-driven, hypothesis-free methods such as PCA. This is because the actual dimensionality of the DV is effectively one—a scalar function of time that connects the visual evidence to a saccadic choice (*Ganguli et al., 2008*). Thus a weighting established by training a decoder at time $t = \tau$ to predict the monkey's choice performs nearly as well when tested at times other than the time the decoder was trained on (i.e., $t \neq \tau$; *Figure 4*).

The different strategies for deriving coding directions lead to different weight assignments, but the coding directions are linearly dependent (*Figure 4d*). They produce traces, $S(t)$, that are similar (*Figure 4e*) and suggestive of drift-diffusion. Traces accompanying trials with the same motion coherence meander and spread apart at a rate similar to diffusion (i.e., standard deviation proportional to $\sqrt{t}$), and they exhibit a pattern of autocorrelation, as a function of time and lag, consistent with diffusion (*Figure 3*). The calculations applied in the present study improve upon previous applications (e.g., *Churchland et al., 2011*; *de Lafuente et al., 2015*; *Shushruth et al., 2018*) by incorporating the contribution of the smoothing to the autocorrelations. The departures from theory are explained by the fact that the accumulations are bounded. The upper bound and the fact that spike rates must be non-negative (a de facto lower reflecting bound) limits the spread of the single-trial traces.

The single-trial signals, $S_i^x(t)$, approximate the DV that gives rise to the choice and decision time on trial $i$ (*Figure 5*, *Figure 5—figure supplement 3*). Support for this assertion is obtained using a conservative assay, which quantifies the leverage of the first 300 ms of the signal's evolution on decision outcomes—choice and RT—occurring at least 670 ms after motion onset. Naturally, the signals do not explain all the variance of these outcomes. The sample size is limited to $N$ randomly selected, often weakly correlated neurons. The sample size and correlation are especially limiting for the $\text{T}_{\text{in}}^{\text{con}}$ neurons ($\mathbb{E}(r) = 0.067 \pm 0.0036$). Control analyses show that the degree of leverage on behavior and mediation of these relationships by later activity is on par with that obtained from simulated, weakly correlated neurons (*Figure 5—figure supplement 1*). In addition, because they are identified post hoc, many have response fields that barely overlap the choice target. Presumably, that is why their responses are weak compared to previous single-neuron studies in which the choice targets were centered in the response field by the experimenter. Yet even this noisy signal, $S_{\text{Tin}}^{\text{con}}$, mediates signals produced by coding directions using the entire population (*Figure 5*).

The $T_{in}^{con}$ neurons were the first to be identified as a plausible candidate neural representation of the DV, based on firing rate averages (*Shadlen and Newsome, 1996*; *Platt and Glimcher, 1999*). This neural type is also representative of the LIP projection to the region of the SC that represents the saccadic vector required to center the gaze on the choice target. (*Paré and Wurtz, 1997*). In a companion study by *Stine et al., 2023* we show that the SC is responsible for cessation of integration in LIP. Features of the drift-diffusion signal from the $T_{in}^{con}$ neurons are correlated with bursting events in corresponding populations of $T_{in}^{con}$ neurons in the SC, including the final saccadic burst that ends the decision with a saccade to the contralateral choice target. *Stine et al., 2023* also show that inactivation of $T_{in}^{con}$ neurons in the SC has little effect on $S_{Tin}^{con}$ signals in LIP.

Previous studies of LIP using the random dot motion task focused primarily on the $T_{in}^{con}$ neurons (*cf. Meister et al., 2013*). It was thus unknown whether and how other neurons contribute to the decision process. The Neuropixels probes used in the present study yield a large and unbiased sample of neurons. Many of these neurons have response fields that overlap one of the two choice targets, but the majority have response fields that overlap neither the choice targets nor the random dot motion. Our screening procedures (delayed saccades and passive motion viewing tasks) do not supply a quantitative estimate of their spatial distribution. It is worth noting that neurons with response fields that overlap neither of the two choice targets were assigned nonzero weights by the *What-* and *When*-decoders, and yet, removal of the task-related neurons that represent the choice targets and motion (i.e., $T_{in}$ and $M_{in}$) decreases decoding accuracy more substantially than removing all but the $T_{in}$ neurons (*Figure 4—figure supplement 5*). The accuracy the decoder achieves is likely explained by neurons with weak responses that simply failed to meet our criterion for inclusion in the $T_{in}$ and $M_{in}$ categories (e.g., neurons with response fields that barely overlap the choice targets or RDM). Some neurons outside these groups might reflect normalization signals from the $T_{in}$ and $M_{in}$ neurons (*Shushruth et al., 2018*; *Carandini and Heeger, 2011*), imbuing broad, decision-related co-variability across the population. It thus seems possible that higher dimensional tasks (e.g., four choices instead of two) could decrease correlations among groups of neurons with different response fields.

The fact that the raw averages from a small number of weakly correlated $T_{in}$ neurons furnish a DV on par with that furnished by the full population underscores the importance of this functional class. The role of the $M_{in}$ neurons is less well understood. Freedman and colleagues described direction selective neurons in LIP, similar to our $M_{in}$ neurons (*Freedman and Assad, 2011*; *Fanini and Assad, 2009*; *Sarma et al., 2016*). They showed that the neurons represent both the direction of motion and the decision in their task. In contrast, we do not observe the evolution of the decision (i.e., DV) by the $M_{in}$ neurons (*Figure 6*). The latency of the direction and coherence-dependent signal as well as its dynamics resemble properties of DS neurons in area MT. The delayed correlation between $M_{in}$ and $T_{in}$ responses evokes the intriguing possibility that $M_{in}$ neurons supply the momentary evidence, which is integrated within LIP itself (*Zhang et al., 2022*). Future experiments that better optimize the yield of $M_{in}$ neurons will be informative, and direct, causal support will require perturbations of functionally identified $M_{in}$ neurons, which is not yet feasible. A natural question is why LIP would contain a copy of the DS signals that are already present in area MT. We suspect it simplifies the routing of momentary evidence from neurons in MT/MST to the appropriate $T_{in}$ neurons. This interpretation leads to the prediction that DS $M_{in}$ neurons would be absent in LIP of monkeys that are naïve to saccadic decisions informed by random dot motion, as has been observed in the SC (*Horwitz et al., 2004*). Further, when motion is not the feature that informs the saccadic response—for example, in a color categorization task (e.g., *Kang et al., 2021*)—LIP might contain a representation of momentary evidence for color (*Toth and Assad, 2002*; *Sereno and Maunsell, 1998*).

The capacity to record from many neurons simultaneously invites characterization of the population in NSS, in which the activity of each neuron defines a dimension. Often, population activity is confined to a low-dimensional subspace or manifold within the NSS (*Vyas et al., 2020*). An ever-more-popular viewpoint is that representations within these subspaces are emergent properties of the population—that is, *distributed*, rather than coded *directly* by single neurons—a dichotomy that has its roots in Barlow's neuron doctrine (as updated in *Barlow, 1994*). Indeed, it is tempting to conclude that the drift-diffusion signal in LIP is similarly emergent based on our NSS analyses—the identified subspaces (i.e., coding directions) combine neurons with highly diverse activity profiles. In contrast, grouping neurons by the location of their spatial response field reveals a direct coding scheme: $T_{in}$ neurons directly represent the accumulated evidence for making a particular saccade and $M_{in}$ neurons

represent the momentary evidence. We argue that this explanation is more parsimonious and, importantly, more principled. Grouping neurons based on spatial selectivity rests on the principle that neurons with similar RFs have similar projections, which is the basis for topographic maps in the visual and oculomotor systems (*Schall, 1995*; *Silver and Kastner, 2009*; *Kremkow et al., 2016*; *Felleman and Van Essen, 1991*). In contrast, there are no principles that guide the grouping of neurons in state space analyses, as the idea is that they may comprise as many dimensions as there are neurons that happen to be sampled by the recording device.

The present finding invites both hope and caution. It may be useful to consider a counterfactual state of the scientific literature that lacks knowledge of the properties of LIP $T_{in}$ neurons—a world without *Gnadt and Andersen, 1988* and no knowledge of LIP neurons with spatially selective persistent activity. In this world we have no reason to entertain the hypothesis that decisions would involve neurons that represent the choice targets. We do know about DS neurons in area MT and their causal role in decisions about the direction of random dot motion (*Salzman et al., 1992*; *Fetsch et al., 2018*; *Ditterich et al., 2003*; *Liu and Pack, 2017*). We also know that drift-diffusion models explain the choice-response time behavior. Guided by no particular hypothesis, we obtain population neural recordings in the random dot motion task. We do not perform the saccade and passive viewing control experiments. What might we learn from such a dataset? We might apply PCA and/or train a choice decoder or possibly a *When*-decoder. If so, we could discover the drift-diffusion signal and we might also infer that the dimensionality of the signal is low. However, we would not discover the $T_{in}$ neurons without a hypothesis and a test thereof. We might notice that the coding directions that reveal drift-diffusion often render a response at the onset of the choice targets as well as increased activity at the time of saccades to the contralateral choice target. These facts might lead us to hypothesize that the population might contain neurons with visual receptive fields and some relationship to saccadic eye movements. We might then query individual neurons, post hoc, for these features, and ask if they render the drift-diffusion signal too. The inferences could then be tested experimentally by including simple delayed saccades in the next experiment. The hope in this counterfactual is that data-driven, hypothesis-free methods can inspire hypotheses about the mechanism. The caution is to avoid the natural tendency to stop before the hypotheses and tests, thus accepting as an endpoint the characterization of population dynamics in high dimensions or a lower dimensional manifold. If LIP is representative, these mathematically accurate characterizations may fail to illuminate the neurobiological parsimony.

## Methods

### Ethical approval declarations

Two adult male rhesus monkeys (*Macaca mulatta*, Primate Products) were used in the experiments. All training, surgery, and experimental procedures complied with guidelines from the National Institutes of Health and were approved by the Institutional Animal Care and Use Committee at Columbia University (protocols AAAN4900 and AC-AAAW4454). A head post and two recording chambers were implanted under general anesthesia using sterile surgical procedures (for additional details, see *So and Shadlen, 2022*). One recording chamber allowed access to area LIP in the right hemisphere. The other was placed on the midline, allowing access to the SC. Those recordings are described in *Stine et al., 2023*. Here we report only on the neural recordings from LIP, focusing on the epoch of decision formation.

### Behavioral tasks

The monkeys were trained to interact with visual stimuli presented on a CRT video monitor (Vision Master 1451, Iiyama; viewing distance 57 cm; frame rate 75 Hz) using the psychophysics toolbox (*Brainard, 1997*; *Pelli, 1997*; *Kleiner et al., 2007*). Task events were controlled by Rex software (*Hays et al., 1982*). The monkeys were trained to control their gaze and make saccadic eye movements to peripheral targets to receive a liquid reward (juice). The direction of gaze was monitored by an infrared camera (EyeLink 1000; SR Research, Ottawa, Canada; 1 kHz sampling rate). The tasks involve stages separated by random delays, distributed as truncated exponential distributions

$$f(t)=\begin{cases} \dfrac{\alpha}{\lambda}e^{-\frac{t-t_{min}}{\lambda}} & t_{min} \leq t \leq t_{max} \\ 0 & \text{otherwise} \end{cases} \tag{1}$$

where $t_{min}$ and $t_{max}$ define the range, $\lambda$ is the time constant, and $\alpha$ is chosen to ensure the total probability is unity. Below, we provide the range ($t_{min}$ to $t_{max}$) and the exponential parameter $\lambda$ for all variable delays. Note that because of truncation, the expectation $\mathbb{E}(t) < t_{min} + \lambda$.

In the *main task* (*Figure 1a*), the monkey must decide the net direction of random dot motion and indicate its decision when ready by making a saccadic eye movement to the corresponding choice target. After acquiring a central fixation point and a random delay (0.25–0.7 s, $\lambda = 0.15$), two red choice targets (diameter 1 dva) appear in the left and right visual fields. The random dot motion is then displayed after a random delay (0.25–0.7 s, $\lambda = 0.4$ s) and continues until the monkey breaks fixation. The dots are confined to a circular aperture (diameter 5 dva; degrees visual angle) centered on the fixation point (dot density 16.7 dots·dva$^{-2}$s$^{-1}$). The direction and strength of motion are determined pseudorandomly from $\pm\{0, 3.2, 6.4, 12.6, 25.6, 51.2\}$% coherence. The sign of the coherence indicates direction (positive for leftward, which is contraversive with respect to the recorded hemisphere). The absolute value of coherence determines the probability that a dot plotted on frame $n$ will be displaced by $\Delta x$ on frame $n + 3$ ($\Delta t = 40$ms), as opposed to randomly replaced, where $\Delta x = $ dva, consistent with 5 dva·s$^{-1}$ speed of apparent motion (see also *Roitman and Shadlen, 2002*). The monkey is rewarded for making a saccadic eye movement to the appropriate choice target. On trials with 0% coherent motion, either saccadic choice is rewarded with probability 1/2. Errors are punished by extending the intertrial interval by up to 3 s (see *Stine et al., 2023*, for additional details). On approximately half of the trials, a 100 ms pulse of weak motion (±3.2% or 4.0% coherence for monkeys J and M, respectively) is added to the random dot motion stimulus at a random time (0.1–0.8 s, $\lambda = 0.4$) relative to motion onset (similar to *Kiani et al., 2008*). Monkey M performed 9684 trials (five sessions); monkey J performed 8142 trials (three sessions). The data are also analyzed in a companion paper that focuses on the termination of the decision (*Stine et al., 2023*).

In the *visually instructed delayed saccade task* (*Hikosaka and Wurtz, 1983*), one target is displayed at a pseudorandom location in the visual field. After a variable delay (monkey M: 0.4–1.1 s, $\lambda = 0.3$; monkey J: 0.5–1.5 s, $\lambda = 0.2$), the fixation point is extinguished, signaling 'go'. The monkey is rewarded for making a saccade to within ±2.5 dva of the location of the target. In a *memory-guided* variant of the task (*Gnadt and Andersen, 1988*; *Funahashi et al., 1989*), the target is flashed briefly (200 ms) and the monkey is required to make saccade to the remembered target location when the fixation point is extinguished. These tasks provide a rough characterization of the neural response fields during the visual, perisaccadic and delay epochs. Neurons are designated $T_{in}^{con}$ if they exhibit spatially selective activity at the location of the response target in the visual hemifield contralateral to the recorded hemisphere. This determination is made before analyzing the activity in the random dot motion task. We refer to the unweighted mean firing rate as $S_{Tin}^{con}$. Neurons are designated $T_{in}^{ips}$ if they exhibit spatially selective activity at the location of the response target in the visual hemifield ipsilateral to the recorded hemisphere. These analyses were conducted post hoc, after spike sorting.

The *passive motion-viewing task* is identical to the main task, except there are no choice targets and only the strongest motion strength (±51.2% coherence) is displayed for 500 ms (1 s on a small fraction of trials in session 1). The direction is left or right, determined randomly on each trial ($P_{\text{left}} = \frac{1}{2}$). The monkey is rewarded for maintaining fixation until the random dot motion is extinguished.

## Behavioral analyses

We fit a neurally inspired variant of the drift-diffusion model (*Figure 1c*) to the choice-RT data from each session. The model constructs the decision process as a race between two accumulators: one accumulating evidence for left and against right (e.g., left minus right) and one accumulating evidence for right and against left (e.g., right minus left). The decision is determined by the accumulator that first exceeds its positive decision bound, at which point the decision is terminated. The races are negatively correlated with one another, owing to the common source of noisy evidence. We assume they share half the variance, $\rho = -\sqrt{0.5} \approx -0.71$, but the results are robust to a wide range of reasonable values. The decision bounds are allowed to collapse linearly as a function of time, such that

$$B(t) = B_0 - \alpha t, \quad \alpha \geq 0 \tag{2}$$

We used the method of images (*van den Berg et al., 2016*; *Shan et al., 2019*) to compute the probability density of the accumulated evidence for each accumulator (which both start at zero at $t = 0$) as a function of time ($t$) using a time step of 1 ms. The decision time distributions rendered by the model were convolved with a Gaussian distribution of the non-decision times, $t_{nd}$, which combines sensory and motor delays, to generate the predicted RT distributions. The model has six parameters: $\kappa$, $B_0$, $\alpha$, $\mu_{nd}$, $\sigma_{nd}$, and $C_0$, where $\kappa$ determines the scaling of motion strength to drift rate, $C_0$ implements bias in units of signed coherence (*Hanks et al., 2011*), $\mu_{nd}$ is the mean non-decision time, and $\sigma_{nd}$ is its standard deviation (*Table 2*). Additional details about the model and the fitting procedure are described in *van den Berg et al., 2016*.

## Simulated decision variables

We fit the race model described above to the combined behavioral data across all sessions (separately for each monkeys) and used the best-fitting parameters for monkey M (see *Table 2*) to simulate a total of 60,000 trials representing all signed coherences of the motion discrimination task. Each simulated trial yields a time series for two DVs, one for each accumulator in the race. We assume that the model-derived non-decision time ($t_{nd} = 317$ ms; *Figure 1b*) comprises visual and motor processing times at the beginning and end of the decision: 200 ms from motion onset to the beginning of evidence integration, and the remaining 117ms after termination. The latter approximates the variability observed in the saccadic latencies in the delayed saccade task and is simulated using a normal distribution, $\mathcal{N}(\mu, \sigma)$, where $\mu = 117$ ms and $\sigma = 39$ ms (*Stine et al., 2023*). In this variable time period between decision termination and the response (saccade), the simulated DVs were assigned the values they had attained at the start of this epoch. For all analyses that employ these simulations, we use the DV for the left-choice accumulator because the neural recordings were from LIP in the right hemisphere.

## Neurophysiology

We used prototype 'alpha' version Neuropixels1.0-NHP45 probes (IMEC/HHMI-Janelia) to record the activity of multiple isolated single units from the ventral subdivision of area LIP (LIP$_v$; *Lewis and Van Essen, 2000*). We used anatomical MRI to identify LIP$_v$ and confirmed its physiological hallmarks with single-neuron recordings (Thomas Recording GmbH) before proceeding to multi-neuron recordings. Neuropixels probes enable recording from 384 out of 4416 total electrical contacts distributed along the 45-mm-long shank. All data presented here were recorded using the 384 contacts closest to the tip of the probe (Bank 0), spanning 3.84 mm. Reference and ground signals were directly connected to each other and to the monkey's head post. A total of 1084 neurons were recorded over eight sessions (54–203 neurons per session) (*Table 1*).

The Neuropixels 1.0-NHP45 probe uses a standard Neuropixels 1.0 headstage and is connected via the standard Neuropixels1.0 5m cable to the PCI eXtensions for Instrumentation (PXIe) hardware (PXIe-1071 chassis and PXI-6141 and PXIe-8381 I/O modules, National Instruments). Raw data were acquired using the SpikeGLX software (http://billkarsh.github.io/SpikeGLX/), and single units were identified offline using the Kilosort 2.0 algorithm (*Pachitariu et al., 2016*; *Pachitariu et al., 2020*), followed by manual curation using Phy (https://github.com/cortex-lab/phy).

## Neural data analysis

The spike times from each neuron are represented as delta functions of discrete time, $s_{i,n}(t)$, on each trial $i$ and each neuron $n$ ($dt = 1$ ms). The weighted sum of these $s_{i,n}(t)$ gives rise to the single-trial population signals:

$$S_i^x(t) = \sum_n w_n s_{i,n}(t) \tag{3}$$

where the superscript, $x$, identifies the method or source that establishes the weights—that is, the coding direction in NSS or the neuron type contributing to a pooled average (e.g., $T_{in}^{con}$). For visualization, the single-trial signals are smoothed by convolution with a truncated Gaussian using the MATLAB function, *gausswin* (width = 80 ms, width factor = 1.5, $\sigma \approx 26$ ms). Unless otherwise specified, all other analyses employ a 50 ms boxcar (rectangular) filter; values plotted at time $t$ include data from $t - 24$ to $t + 25$ ms.

We used several methods to define coding directions in the NSS defined by the population of neurons in each session. For PCA and choice decoding, we standardized the single-trial firing rates for each neuron using the mean and standard deviation of its firing rate at in the epoch $200 \leq t \leq 600$ ms after motion onset. This practice led to the exclusion of two neurons (session 1) that did not produce any spikes in the normalization window. Those neurons were assigned zero weight.

### $T_{in}$ neurons

Neurons were classified post hoc as $T_{in}$ by visual-inspection of spatial heatmaps of neural activity acquired in the delayed saccade task. We inspected activity in the visual, delay, and perisaccadic epochs of the task. The distribution of target locations was guided by the spatial selectivity of simultaneously recorded neurons in the SC (see *Stine et al., 2023*, for details). Briefly, after identifying the location of the SC response fields, we randomly presented saccade targets within this location and seven other, equally spaced locations at the same eccentricity. In monkey J, we also included 1–3 additional eccentricities, spanning 5–16 degrees. Neurons were classified as $T_{in}$ if they displayed a clear, spatially selective response in at least one epoch to one of the two locations occupied by the choice targets in the main task. Neurons that switched their spatial selectivity in different epochs were not classified as $T_{in}$. The classification was conducted before the analyses of activity in the motion discrimination task. The procedure was meant to mimic those used in earlier single-neuron studies of LIP (e.g., *Roitman and Shadlen, 2002*) in which the location of the choice targets was determined online by the qualitative spatial selectivity of the neuron under study. The $T_{in}^{con}$ neurons in the present study were highly selective for either the contralateral or ipsilateral choice target used in the RDM task (AUC $= 0.89 \pm 0.01$ for 97% of neurons, Wilcoxon rank-sum test). Given the sparse sampling of saccade target locations, we are unable to supply a quantitative estimate of the center and spatial extent of the RFs. We next describe the methods to establish the coding directions.

### Ramp direction

We applied linear regression to generate a signal that best approximates a linear ramp, on each trial, $i$, that terminates with a saccade to the choice target contralateral to the hemisphere of the LIP recordings. The ramps are defined in the epoch spanning the decision time: each ramp begins at $f_i(t_0) = -1$, where $t_0 = 0.2$ s after motion onset, and ends at $f_i(t_1) = 1$, where $t_1 = t_{sac} - 0.05$ s (i.e., 50 ms before saccade initiation). The ramps are sampled every 25 ms and concatenated using all eligible trials to construct a long saw-tooth function (see *Figure 2—figure supplement 1*). The regression solves for the weights assigned to each neuron such that the weighted sum of the activity of all neurons best approximates the saw-tooth. We constructed a time series of standardized neural activity, sampled identically to the saw-tooth. The spike times from each neuron are represented as delta functions (rasters) and convolved with a non-causal 25 ms boxcar filter. The mean and standard deviation of all sampled values of activity were used to standardize the activity for each neuron (i.e., Z-transform). The coefficients derived from the regression establish the vector of weights that define $S^{ramp}$. The algorithm ensures that the population signal $S^{ramp}(t)$, but not necessarily individual neurons, have amplitudes ranging from approximately -1 to 1.

We employed a lasso linear regression with $\lambda = 0.005$. The vector of weights assigned across the neurons defines a direction in NSS, $S^{ramp}$, which we use to render the signal $S^{ramp}(t)$ on single trials by projecting the data onto this direction. To determine the effect of the regularization term in the lasso regression, we recomputed single-trial signals using standard linear regression, without regularization. We then calculated the Pearson correlation between single-trial traces generated by projecting neural data onto the two coding directions (i.e., with and without regularization). The high correlation between single-trial traces (mean $r = 0.99$, across sessions) indicates that the findings are not a result of the regularization applied. Here and elsewhere we compute the mean $r$ using the Fisher-z transform, such that

$$\bar{r} = Z^{inv}\left(\mathbb{E}\left[Z(r_k)\right]\right) \tag{4}$$

where $Z^{inv}$ is the inverse Fisher-z.

### Principal component analysis (PCA)

We applied standard PCA to the firing rate averages for each neuron using all trials sharing the same signed motion coherence in the shorter of two epochs: 200 ms to either 600 ms after motion onset or

100 ms before the median RT for the signed coherence, whichever produces the shorter interval. The results of the PCA indicate that the dimensionality of the data is low, but greater than 1. The participation ratio is 4.4 ± 1.3 (*Mazzucato et al., 2016*; *Gao et al., 2017*) and the first three PCs explain 67.1 ± 3.1% of the variance on average (mean ± s.e.m. across sessions). As in all other analyses of neural activity aligned to motion onset, we exclude data in the 100 ms epoch ending at saccade initiation on each trial. We projected the neural data onto the first PC to generate the signal $S^{\mathrm{PC1}}$.

### Choice decoder

For each experimental session, we trained logistic choice decoders with lasso regularization ($\lambda = 0.01$) on the population activity in 50 ms time bins spanning the first 500 ms after motion onset and the 300 ms epoch ending at saccade initiation, respectively. Each of the decoders was trained on the even-numbered trials. Decoder accuracy was cross-validated using the activity of held-out, odd trials at the same time point (*Figure 4a*). For the time bins aligned to motion onset, we also assessed the accuracy of the decoders trained on each of the time bins to predict the choice on time bins on which they were not trained (*Figure 4b*; *King and Dehaene, 2014*). We use the decoder trained on the bin centered on $t = 450$ ms to define the *What* coding direction. We refer to this as the *fixed training-time* decoder to distinguish it from the standard machine-learning decoder, which assigns a potentially distinct vector of weights at each time point. We applied a similar analysis to simulated data (see 'Simulated DVs') to generate the black curve in *Figure 4a*. Assuming a stochastic drift-diffusion process giving rise to choice and response times, the exercise establishes a rough upper bound on decoder accuracy, were the actual drift-diffusion process known precisely.

### When decoder

This decoder is trained to predict whether a saccade to the left (contralateral) choice target will occur within the next 150 ms. We applied logistic regression with lasso regularization ($\lambda = 0.01$) to spike counts from each neuron in discrete bins of 25 ms, from 200 ms before motion onset to 50 ms before the saccade. We used only trials ending in a left choice (including errors) and trained the decoder on the even numbered half of those trials. The concatenation of these trials forms a sequence of step functions which are set to 1 if a saccade occurred within 150 ms of the start of the 25 ms time bin and 0 otherwise (*Figure 4—figure supplement 1*).

The spike counts were also concatenated across these trials to construct column vectors (one per neuron) that match the vector of concatenated step functions. These concatenated vectors, one per neuron, plus an offset ($\beta_0$), serve as the independent variables of the regression model (one $\beta$ term per neuron). The proportion of $\beta$ weights equal to zero, controlled by the lasso parameter, $\lambda$, was $0.8 \pm 0.02$ across sessions. The weights define the $S^{\mathrm{When}}$ coding direction, which yields single-trial signals, $S^{\mathrm{When}}(t)$. The When-decoder signal is $S^{\mathrm{When}}(t) + \beta_0$. We validated the When-decoder by computing the area under an ROC (AUC) using the held-out (odd-numbered) trials ending in left choices (mean AUC over all time points and sessions: $0.84 \pm 0.024$, mean ± s.e.).

Our motivation, however, was to ascertain whether the *When* coding direction also predicts the monkey's choices on *all trials*—that is, to perform as a *What* decoder. To this end, we predicted the choice using the sign of the detrended $S^{\mathrm{When}}(t)$, formed by subtracting the average of the signal using all trials:

$$S_i^*(t) = S_i^{\mathrm{When}}(t) - \langle S_i^{\mathrm{When}}(t) \rangle_i \tag{5}$$

where $\langle \cdots \rangle_i$ denotes expectation across all trials contributing values at time $t$. The choice accuracy is

$$A_i(t) = \begin{cases} 1 & S_i^*(t) > 0 \ \&\ \mathrm{choice}_i = \mathrm{left} \\ 1 & S_i^*(t) \leq 0 \ \&\ \mathrm{choice}_i = \mathrm{right} \\ 0 & \mathrm{otherwise} \end{cases}$$

The green trace in *Figure 4a* shows $\langle A_i(t) \rangle_i$.

### Aggregation of data across experimental sessions

To combine single-trial data across sessions (e.g., $S^x(t)$), we first normalize activity within each session as follows. Using all trials ending in the same choice, we construct the trial-averaged activity aligned

to both motion onset ($0 \leq t_{motion} \leq 0.6$ s) and saccade onset ($-0.6 \leq t_{sacc} \leq 0$ s). This produces four traces. The minimum and maximum values ($a_{min}$ and $a_{max}$) over all four traces establish the range, zero to one, of the normalized signal:

$$\hat{s}^x(i,t) = \frac{s^x(i,t) - a_{min}}{a_{max} - a_{min}} \tag{6}$$

where lowercase $\hat{s}^x(i,t)$ is the normalized signal across trials $i$ and time $t$ in an individual session.

## Cosine similarity

We computed the cosine similarity between the weight vectors that define coding directions $S^{\text{ramp}}$, $S^{\text{PC1}}$, $S^{\text{con}}_{\text{Tin}}$, $S^{\text{What}}$, and $S^{\text{When}}$. Mean cosine similarities are portrayed in the heatmap in *Figure 4d* and also in *Figure 4—figure supplement 3*, top, where they are accompanied by error bars. We evaluated the null hypothesis that the mean cosine similarity is ≤0 with *t*-tests. We also performed two control analyses that deploy random coding directions in NSS. For each of the original coding directions, we obtained 1000 random coding directions as random permutations of the weight assignments. The cosine similarities between pairs of such random directions in state space are shown in *Figure 4—figure supplement 3*, top. The cumulative distribution of cosine similarities under permutation supports p-values less than 1/1000. In a second control analysis, we used random unit vectors as random coding directions (normal distribution with mean 0 and scaled to unit length).

## Similarity of single-trial signals

We calculated Pearson correlation to quantify the similarity of the signals generated by pairs of coding directions, $x$ and $y$. For each trial, $i$, the detrended signals, $\widetilde{S}^x_i$ and $\widetilde{S}^y_i$, provide ordered pairs, $\{\widetilde{S}^x_i(t_j), \widetilde{S}^y_i(t_j)\}$ where $j$ indexes successive 50 ms bins between 200 ms after motion onset and 100 ms before saccade initiation. We excluded trials comprising less than four such bins. Each trial gives rise to a correlation coefficient, $r_i$. We report the mean $r$ using *Equation 4*. The $r$-values for comparisons across all pairs of coding directions are summarized in *Figure 4e*, and variability across sessions is portrayed in *Figure 4—figure supplement 3*, bottom. For each pair of CDs and session, we evaluated the null hypothesis, $\mathcal{H}_0 : \bar{r} \leq 0$ (*t*-test).

We also performed two control analyses that deploy random CDs in NSS. These analyses control for the possibility that the correlations observed in the signals are explained by pairwise correlations between the neurons, regardless of the signals produced by the weighted sums. (i) We generated sets of single-trial traces $S^{\text{rand}}(t)$ by projecting the neural responses onto random CDs, defined by permuting the weights of each coding direction ($S^{\text{ramp}}$, $S^{\text{PC1}}$, $S^{\text{con}}_{\text{Tin}}$, $S^{\text{What}}$, and $S^{\text{When}}$). For each pair of CDs, we compute within-trial correlations between ordered pairs of trials using the same method applied to the original signals. We repeat this process for a total of 1000 random permutations per pair of CDs, per session. (ii) We sample a pair of random weight vectors from standard normal distributions. Each weight vector has a dimension equal to the number of recorded neurons in the session. The weight vectors are normalized to sum to 1. We generate random CDs using these weights and compute within-trial correlations using the same method applied to the original signals. We repeat this process 1000 times per session. For both analyses, we evaluated the null hypothesis that the observed correlations are not greater than those produced by the random projections (*t*-tests using the Fisher-z transformed correlations). Mean ± stdev of the mean r-values between ordered pairs for both control analyses are summarized in *Figure 4—figure supplement 3*, bottom.

## Leverage of single-trial activity on behavior

The leverage of single-trial signals, $S^x(t)$, on choice and RT was assessed using the earliest 300 ms epoch of putative integration ($0.2 < t < 0.5$ s from motion onset), restricting analyses to trials with RTs outside this range ($0.67 < \text{RT} < 2$ s). The single-trial signals are smoothed with a 50 ms boxcar filter and detrended by subtracting the mean $S(t)$ for trials sharing the same motion strength and direction (i.e., signed coherence). The RTs are also expressed as residuals relative to the mean RT, using trials sharing the same signed coherence and choice. We include trials with |coh| ≤ 0.064 that result in choices of the left response target in this analysis. Including trials of |coh| ≤ 0.128 produced comparable results. The leverage on RT is the Pearson correlation between the residual signals $\widetilde{S}(t)$ at each time $0.2 < t \leq 0.5$, and $\widetilde{RT}$ on that trial, where the tilde indicates residual. Correlations were

computed per session and then averaged across sessions (*Equation 4*). We also show the correlation at $t = 0.55$ s, using the ordered pairs, $\{\tilde{S}(t = 0.55), \tilde{RT}\}$. We quantify mediation of the leverage of earlier samples by the later sample of $S^x$ by computing partial correlations $\{\tilde{S}(t), \tilde{RT} \mid \tilde{S}(t = 0.55)\}$, also notated $R_{\tilde{S}(t),\tilde{RT}|\tilde{S}(0.55)}$ in *Figure 5*. We show this mediation at all time points. We also report a mediation statistic ($\xi^{RT}$) using the time point 200 ms after the beginning of putative integration (i.e., $S(t = 0.4)$):

$$\xi^{RT} = 1 - \frac{R^2_{\tilde{S}(0.4),\ \tilde{RT} \mid \tilde{S}(0.55)}}{R^2_{\tilde{S}(0.4),\ \tilde{RT}}} \tag{7}$$

The rationale for using the 400 ms time point is (i) to allow the process to have achieved enough leverage on RT so that a reduction is meaningful and (ii) to preserve a substantial gap between this time and the sample at $t = 0.55$ s (e.g., to avoid autocorrelations imposed by smoothing). The rare cases in which there was no negative correlation between $S(t = 0.4)$ and RT were excluded from this summary statistic, $\xi^{RT}$, because no mediation is possible (Session 2: $S^{ramp}$ & $S^{PC1}$). When combining values of $\xi^{RT}$ across sessions, we rectify any $R_{\tilde{S}(t),\tilde{RT}|\tilde{S}(0.55)} > 0$ to zero. This occurs rarely when the mediated correlation is near zero, typically at early times.

We compute the leverage on choice, $\xi^{Ch}$, using trials with $|coh| \leq 0.064$ and the same time points as for $\xi^{RT}$. Instead of an R-squared measure, we based $\xi^{Ch}$ on coefficients derived from logistic regression:

$$P^{con}(t) = \frac{1}{1 + e^{-(\beta_0^{coh} + \beta_1(t)S(t))}}, \quad 0.2 \leq t \leq 0.5. \tag{8}$$

where $\beta_0^{coh}$ is a set of constants that accounts for the proportion of contralateral choices at each signed motion strength and $\beta_1(t)$ is the simple leverage of $S(t)$ on choice, analogous to simple correlation. The regression analysis was performed separately for each session. The coefficients $\beta_1(t)$ were divided by their standard error and then averaged across sessions. This normalization step was implemented to control for potential variation in the magnitude of $S(t)$ (and therefore of $\beta_1(t)$) across sessions. Analogous to partial correlation, we include the later time point and fit

$$P^{con}(t) = \frac{1}{1 + e^{-(\beta_0^{coh} + \beta_1^*(t)S(t) + \beta_2 S(0.55))}} \tag{9}$$

where $\beta_1^*(t)$ is the amount of leverage at time $t$ given the $S(0.55)$. The regression coefficients $\beta_1^*(t)$ were averaged across sessions after dividing them by the standard error of the $\beta_1(t)$ coefficients obtained from *Equation 8*. That is, the same normalization factors were used for the mediated and unmediated leverage. The summary statistic for choice mediation is defined by

$$\xi^{Ch}(t) = \begin{cases} 1 & \beta_1^*(t) \leq 0 \\ 1 - \dfrac{\beta_1^*(t)}{\beta_1(t)} & \beta_1^*(t) > 0 \\ \text{undefined} & \beta_1(t) \leq 0 \text{ (i.e., mediation is not possible)} \end{cases} \tag{10}$$

For both types of mediation, we also test whether the earlier $S(t)$ is mediated by the $T_{in}^{con}$ neurons by substituting $S_{T_{in}}^{con}(0.55)$ for $S(0.55)$ in *Equation 9* and in the expression for partial correlations.

Because the mediation statistics, $\xi^{RT}$ and $\xi^{Ch}$, are, by their definition, non-negative, we assess statistical significance by bootstrapping. For each session, we construct 1000 surrogate data sets equal in size to the original data by sampling with replacement. The standard deviation of the leverage and mediation values at each time approximates the standard error. We compare the distribution of the mediation statistics, $\zeta$, to their distribution under the null hypothesis, $\mathcal{H}_0$, that the values arise by chance, instantiated by breaking the correspondence with the trial giving rise to the later sample, $S^x(t = 0.55s)$. The permutation maintains correspondence in signed motion coherence. We compare the distributions of mediation from the bootstrap and $\mathcal{H}_0$ using the Wilcoxon rank-sum test.

To test whether the observed leverage of neural activity on choice and RT is achieved by projections onto arbitrary coding directions, we generated random weight vectors by permuting the weights associated with the first PC for each session. We projected activity onto this random coding direction,

applied the mediation analyses described above to this signal, and repeated this process 1000 times to produce a null distribution at each time point. The reported p-values represent the probability that the observed leverage was generated from this null distribution.

We performed a similar analysis to test whether the observed leverage depends on the trial-to-trial correspondence between neural activity and behavior. Here, the null distribution at each time point was generated by randomly permuting the trial indices associated with the neural activity and those associated with the behavioral measures.

Finally, we used the simulated data from the racing accumulator model to test the degree of leverage and mediation expected had we known the ground-truth DV on each trial (*Figure 5—figure supplement 1*). Because the simulated DV is noiseless (and the process is Markovian), the mediation is expected to be complete for all time points tested in the analyses. We therefore took two steps to make the simulated data more comparable to neural data: (i) we sub-sampled the simulated data to match the number of trials in each session. (ii) We generated $N$ noisy instantiations of the signal for each of the sub-sampled, simulated trials, where $N$ is the number of $T_{in}^{con}$ neurons in each session. The added noise is independent across time points and weakly correlated across all $N \times (N-1)/2$ neuron pairs ($r \approx 0.09$). We then applied the mediation analyses to the mean of these signals and repeated this process 1000 times.

We performed three control analyses to determine whether the results of the mediation analysis were specific, meaningful, and comparable to the results obtained for the DV of a race model. To showcase that the results are specific, we generated random weight vectors by permuting the weights of the PC1 coding direction. We repeated this procedure 1000 times per session and projected the data along these directions in NSS to generate $S^{rand}$. We then computed the mediation analyses detailed above on these signals and determined significance by comparing the leverage on choice and correlation with RT of $S^{ramp}$, $S^{PC1}$, and $S_{Tin}^{con}$ to the null distributions of $S^{rand}$ at each time point.

To estimate an upper limit for the degree of possible leverage and mediation, we simulated 60,000 trials using the race model that best fits the behavioral data of monkey M (see 'Simulated DVs'). For any noise-free representation of a Markovian integration process, the leverage of an early sample of the DV on behavior would be mediated completely by later activity as the latter sample by definition encompasses all variability captured by the earlier sample. We, therefore, took two steps to make the simulated DVs more comparable to real neural data. (i) For each session, we first subsampled the simulated data to match the each session. (ii) To evaluate a DV approximated from the activity of $n$ neurons per session rather than the true DV represented by the entire population, we then generated $n$ noisy instantiations of the signal for each simulated trial. The added noise is independent across time points and weakly correlated across neurons ($r \approx 0.09$). We then computed the measured DV $S^{sim}$ as the mean activity of these $n$ simulated neurons. We repeated this procedure 1000 times per session. *Figure 5—figure supplement 1*, bottom, displays the mean and standard deviation across permutations of the leverage of $S^{sim}(t)$ on behavior. The simulation results highlight that we would not expect the mediation of the leverage on behavior by a later sample to be complete (i.e., zero mediated leverage for all $t < 0.55$).

## Noise correlation between neurons

The mean pairwise correlation between $T_{in}^{con}$ neurons, reported in 'Results', is based on all pairs of simultaneously recorded $T_{in}^{con}$ neurons in each session and all trials with $RT > 0.5$ s. For each neuron and each trial, we compute the time-averaged activity over the epoch $0.2 \leq t \leq 0.4$. These scalar values are converted to residuals by subtracting the mean (for each neuron) across all trials sharing the same signed motion coherence. The residuals from all eligible trials are concatenated for each neuron to support the calculation of $N \times (N-1)$ Pearson $r$ values, where N is the number of $T_{in}^{con}$ neurons in the session. The mean correlation for all pairs of $T_{in}^{con}$ neurons across all sessions is computed using *Equation 4*.

## Direction-selective neurons

We identified DS neurons ($M_{in}$ neurons) using the passive motion-viewing task (described above). We classified a neuron as $M_{in}$ if it satisfies two criteria. The first criterion is a short-latency response to the onset of random dot motion, which we defined as a fivefold increase in firing rate relative to baseline in the first 80 ms following motion onset and a greater increase in the rate of rise in

activity in the same 80 ms epoch, compared to any rise in activity in the 200 ms preceding motion onset. The second criterion is direction selectivity. We calculated the area under the ROC (AUC) comparing leftward versus rightward for two separate epochs: (i) $0.15 \leq t < 0.3$ s and (ii) $0.3 \leq t < 0.5$ s. Neurons were determined to be DS if the AUC in either epoch exceeded 0.6. We excluded one neuron from this analysis because it switched its direction preference in the two epochs. We also excluded neurons that had previously been classified as $T_{in}$.

In total, 6 of the 152 $T_{in}^{con}$ neurons fall into this group: three per monkey; at most two in a session. Removal of these neurons has negligible effects on the findings as pairs of $S_{Tin}^{con}$ constructed with and without removal are strongly correlated ($r = 0.9867$).

## Latency analysis

We estimated the latency of DS responses using the CUSUM method (*Ellaway, 1978*; *Lorteije et al., 2015*). We employed a receiver operating characteristic (ROC) analysis to estimate the selectivity of each $M_{in}$ neuron to motion direction. The AUC reflects the separation of the distributions of spike counts (100–400 ms after motion onset) on single trials of leftward and rightward motion, respectively. We included only correct trials with response times greater than 450 ms and motion strengths above 10% coherence. For each neuron with AUC > 0.6, we computed the difference in spike counts (25 ms bins) between correct trials featuring leftward and rightward motion. Subsequently, we accumulated these differences over time, following the CUSUM method. The resulting difference is approximately zero before the onset of direction selectivity and then either increases or decreases monotonically, depending on the preferred motion direction. To identify the transition between these two regimes, we fit a dog leg function to the cumulative sum of spikes: a flat line starting at $t_0 = 0$ followed by a linearly increasing component beginning at $t_1 > t_0$. The time of the end of the flat portion (between 0 and 500 ms from motion onset) of the fit was taken as the latency. Estimating latencies based on cumulative sums of spikes helps mitigate the effect of neuronal noise. The fitting step reduces the effect of the number of trials on latency estimates compared to traditional methods that rely on *t*-tests in moving windows.

## Correlations between $M_{in}$ and $T_{in}^{con}$

The analysis of the correlations shown in *Figure 6e* is based on the spike counts of the $M_{in}^{left}$, $M_{in}^{right}$, and $T_{in}^{con}$ neurons calculated in 25 ms windows. We computed residuals by subtracting from each trial and time bin its average over trials of the same signed coherence. The spike count residuals were then z-scored independently for each time bin and session. Trials from different sessions were concatenated, and the baseline activity—last 100 ms before motion onset—was subtracted from each trial. We refer to the resulting signals as $\tilde{S}_{Min}^{left}$, $\tilde{S}_{Min}^{right}$, and $\tilde{S}_{Tin}^{con}$. Trials with response time less than 0.55 s were discarded, and the correlations between the difference, $\tilde{S}_{Min}^{left} - \tilde{S}_{Min}^{right}$, and $\tilde{S}_{Tin}^{con}$, were calculated for all pairs of time steps between 0 and 500 ms (*Figure 6e*). Statistical significance was assessed using permutation tests, as follows. Two regions of interest (ROIs) were defined based on the time from stimulus onset for the $M_{in}$ ($x$) and $T_{in}$ ($y$) dimensions. The first region of interest, $ROI_1$, is characterized by $t_x > 100$, $t_y > 200$, and $t_y > t_x$. According to our hypothesis that the $M_{in}$ neurons represent the momentary evidence integrated by $T_{in}^{con}$ neurons, we anticipated high correlations in this region. The second region of interest, $ROI_2$, is defined by $t_x > 100$, $t_y > 200$, and $t_y < t_x$. If contrary to our hypothesis $M_{in}$ and $T_{in}$ signals were influencing each other bidirectionally, we would expect high correlations in this region. We calculated the difference in correlations between these two groups, $\langle \rho_{ROI1} \rangle - \langle \rho_{ROI2} \rangle$, where the expectation is over the time bins within each region of interest. This difference was compared to those obtained after randomly shuffling the order of the trials for one of the dimensions before calculating the pairwise correlations ($N_{shuffles} = 200$). We assess significance with a *z*-test given the mean and standard deviation of the values obtained under shuffling. The analysis was repeated with an alternative $ROI_2$ defined by $t_x < 100$ and $t_y < 200$, representing the times before direction selectivity is present in at least one of the two dimensions.

## Correlations between $M_{in}$ signals and behavior

To assess the leverage of $M_{in}$ signals on choice and RT (*Figure 6d*), we performed the same logistic regression and pairwise correlation analyses as in *Figure 5*, substituting the $M_{in}^{left}$ and $M_{in}^{right}$ for $S^x$. The leverage on choice is not mediated by a later sample of either $M_{in}$ signal ($\xi^{Ch} \leq 9.6\%$;

not shown), and there is negligible leverage on RT to mediate. We suspect the failure to detect leverage of $M_{in}$ is explained by a lack of power, owing to the focus on long RT trials, narrow sample windows (50 ms boxcar), and the small number of $M_{in}^{left}$ and $M_{in}^{right}$ neurons. We support this suspicion with a simpler correlation analysis using the difference of the $M_{in}$ signals (standardized as in the previous paragraph):

$$\psi_k = \text{mean}\left(\tilde{S}_{Min}^{left}(t) - \tilde{S}_{Min}^{right}(t)\right) \tag{11}$$

on the interval $0.1 \leq t \leq 0.4$ s from motion onset, on each trial, $k$, including trials with contraversive choices and $RT \geq 500$ ms. We calculated the Pearson correlation coefficient between $\psi_k$ and RT. Response times were z-scored independently for each signed motion strength and session. We evaluated the null hypothesis that the correlation coefficient is non-negative. The reported p-value is based on a one-tailed $t$-statistic.

## Variance and autocorrelation of smoothed diffusion signals

The analyses in *Figure 3* compare the variance and autocorrelation of the single-trial signals, $S^{ramp}(t)$, to those expected from unbounded drift-diffusion. To mitigate the effect of the bound, we focus on the earliest epoch of putative integration (200–506 ms after motion onset; six 51 ms counting windows) and the weakest motion strengths (|coh| $\leq 3.2\%$). The single-trial signals are detrended by the mean across trials sharing the same signed motion coherence and baseline corrected by subtraction of $S_i^{ramp}(t = 0.2)$ from all time points on each trial $i$.

The variance as a function of time and the autocorrelation as a function of time and lag are well specified for the cumulative sum of discrete *iid* random samples, but the autocorrelation is affected by the boxcar filter we applied to render the signals. We incorporated the correction in our characterization of unbounded diffusion. The derivation is summarized in Appendix 1, and we provide MATLAB code in the GitHub repository. The theoretical values shown in *Figure 3* assume a 1 kHz sampling rate and standard Wiener process (i.e., samples drawn from a normal distribution with $\{\mu = 0, \sigma = \sqrt{dt}\}$). The evolution of variance would be a line from 0 to 1 over the first second of integration. The key prediction, shown in *Figure 3a*, is that the variance of mean single-trial signals, $S_i(t)$, over the epoch $26 \pm 25$ ms should double in the epoch $(26 + 51) \pm 25$ ms, and triple in the epoch $(26 + 2 \times 51) \pm 25$ ms), and so on for each successive non-overlapping running mean. We therefore use arbitrary units, normalized to the measured variance of the first point. We do not know the variance of the drift-diffusion signal that $S(t)$ is thought to approximate, but we assume it can be decomposed—by the law of total variance—to a component given by drift-diffusion and components associated with spiking and other nuisance factors. We therefore search for a scalar non-negative factor $\phi \leq 1$ that multiplies all terms in the diagonal of the empirical covariance matrix (i.e., the variance) before normalizing to produce the autocorrelation matrix. We search for the value of $\phi$ that minimizes the sum of squares between Fisher-z transformed correlation coefficients in the theoretical and empirical autocorrelation matrices (*Figure 3b and c*). Standard errors of the variance and autocorrelations in *Figure 3a and c* are estimated by a bootstrap procedure respecting the composition of motion strength and direction (the s.e. is standard deviation of each variance and autocorrelation term across 500 repetitions of the procedure.

## Acknowledgements

We thank Shushruth, NaYoung So, and David Gruskin for comments on the manuscript, Cornel Duhaney and Brian Madeira for their assistance in the planning and execution of surgeries, animal training and general support, and we thank Columbia University's ICM for the quality of care they provide for our animals, especially during the pandemic and lockdown. We would further like to thank Tanya Tabachnik and her team at the Zuckerman Institute Advanced Instrumentation Core and Tim Harris, Wei-lung Sun, Jennifer Colonell, and Bill Karsh at HHMI Janelia for their continued support with Neuropixels1.0-NHP45 probes development and testing. This research was supported by the Howard Hughes Medical Institute; an R01 grant from the NIH Brain Initiative (MNS, R01NS113113); a T32 and F31 grant from the National Eye Institute (GMS, T32 EY013933, F31 EY032791); the Grossman center; and the Brain and Behavior Research Foundation. DMW is a consultant to CTRL-Labs Inc, in the Reality Labs Division of Meta. This entity did not support or influence this work.

# Additional information

### Competing interests

Daniel M Wolpert: consultant to CTRL- Labs Inc, in the Reality Labs Division of Meta. This entity did not support or influence this work. The other authors declare that no competing interests exist.

### Funding

| Funder | Grant reference number | Author |
|---|---|---|
| Howard Hughes Medical Institute | | Ariel Zylberberg Michael N Shadlen |
| National Institutes of Health BRAIN Initiative | R01NS113113 | Natalie Steinemann Michael N Shadlen |
| National Eye Institute | T32 EY013933 | Gabriel M Stine |
| Grossman Center | Zuckerman Institute | Eric Trautmann |
| Brain and Behavior Research Foundation | | Eric Trautmann |
| National Eye Institute | F31 EY032791 | Gabriel M Stine |

The funders had no role in study design, data collection and interpretation, or the decision to submit the work for publication.

### Author contributions

Natalie Steinemann, Conceptualization, Data curation, Software, Formal analysis, Validation, Visualization, Methodology, Writing – original draft, Project administration, Writing – review and editing; Gabriel M Stine, Conceptualization, Data curation, Software, Formal analysis, Investigation, Visualization, Methodology, Writing – original draft, Writing – review and editing; Eric Trautmann, Resources, Supervision, Methodology, Writing – review and editing; Ariel Zylberberg, Conceptualization, Data curation, Software, Formal analysis, Supervision, Validation, Visualization, Methodology, Writing – original draft, Writing – review and editing; Daniel M Wolpert, Conceptualization, Software, Formal analysis, Visualization, Methodology, Writing – original draft, Writing – review and editing; Michael N Shadlen, Conceptualization, Resources, Data curation, Software, Formal analysis, Supervision, Funding acquisition, Validation, Visualization, Methodology, Writing – original draft, Project administration, Writing – review and editing

### Author ORCIDs

Gabriel M Stine (iD) https://orcid.org/0000-0003-4906-0461
Ariel Zylberberg (iD) https://orcid.org/0000-0002-2572-4748
Daniel M Wolpert (iD) https://orcid.org/0000-0003-2011-2790
Michael N Shadlen (iD) https://orcid.org/0000-0002-2002-2210

### Ethics

All training, surgery, and experimental procedures complied with guidelines from the National Institutes of Health and were approved by the Institutional Animal Care and Use Committee at Columbia University (protocols AAAN4900 and AC-AAAW4454).

Reviewer #1 (Public Review): https://doi.org/10.7554/eLife.90859.3.sa1
Reviewer #2 (Public Review): https://doi.org/10.7554/eLife.90859.3.sa2
Author response https://doi.org/10.7554/eLife.90859.3.sa3

# Additional files

### Supplementary files
• MDAR checklist

## Data availability

Matlab code for all analyses and graphs are available at GitHub (copy archived at *Steinemann, 2024*). The data are deposited at Zenodo.

The following dataset was generated:

| Author(s) | Year | Dataset title | Dataset URL | Database and Identifier |
|---|---|---|---|---|
| Steinemann NA, Stine GM, Trautmann EM, Zylberberg A, Wolpert DM, Shadlen MN | 2024 | Data for "Direct observation of the neural computations underlying a single decision" | https://doi.org/ 10.5281/zenodo. 13207505 | Zenodo, 10.5281/ zenodo.13207505 |

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

## Appendix 1

We consider a discrete time (sampling interval $dt$) Wiener process with independent random increments $\epsilon_k$ on time step $k$ that are zero-mean noise with variance $\sigma^2(\epsilon_k) = dt$ (i.e., unit variance per second). The accumulated evidence (i.e., decision variable, DV) on time step $p$ is

$$DV_p = \sum_{k=1}^{p} \epsilon_k \tag{12}$$

For such a Wiener process,

$$\text{Cov}\left(DV_p, DV_q\right) = \text{Cov}\left(\sum_{j=1}^{p} \epsilon_j, \sum_{k=1}^{q} \epsilon_k\right) = \sum_{j=1}^{p} \sum_{k=1}^{q} \text{Cov}\left(\epsilon_j, \epsilon_k\right)$$

As the increments $\epsilon_k$ are independent across time, $\text{Cov}(\epsilon_j, \epsilon_k)$ is 0 for $j \neq k$ and $dt$ for $j = k$

$$\text{Cov}\left(DV_p, DV_q\right) = \sum_{j=1}^{\min(p,q)} \text{Cov}\left(\epsilon_j, \epsilon_j\right) = \min(p, q)dt$$

We define the mean DV over a window of $\pm n$ points as

$$MDV_p^n = \frac{1}{2n + 1} \sum_{j=-n}^{n} DV_{p+j} \tag{13}$$

We consider two time points $p < q$ with window $n$ such that there is no overlap and hence $p + n < q - n$

$$\text{Cov}\left(MDV_p^n, MDV_q^n\right) = \text{Cov}\left(\frac{1}{2n+1} \sum_{j=-n}^{n} DV_{p+j}, \frac{1}{2n+1} \sum_{k=-n}^{n} DV_{q+k}\right)$$

$$= \frac{1}{(2n+1)^2} \sum_{j=-n}^{n} \sum_{k=-n}^{n} \text{Cov}\left(DV_{p+j}, DV_{q+k}\right)$$

$$= \frac{dt}{(2n+1)^2} \sum_{j=-n}^{n} \sum_{k=-n}^{n} \min(p+j, q+k)$$

$$= \frac{dt}{(2n+1)^2} \sum_{j=-n}^{n} \sum_{k=-n}^{n} p + j$$

$$= \frac{dt}{(2n+1)^2}(2n+1)^2 p = p \cdot dt$$

$$\text{Var}\left(MDV_p^n\right) = \text{Var}\left(\frac{1}{2n+1} \sum_{j=-n}^{n} DV_{p+j}\right)$$

$$= \frac{1}{(2n+1)^2} \sum_{j=-n}^{n} \sum_{k=-n}^{n} \text{Cov}\left(DV_{p+j}, DV_{p+k}\right)$$

$$= \frac{1}{(2n+1)^2} \sum_{j=1}^{2n+1} \sum_{k=1}^{2n+1} \text{Cov}\left(DV_{p-n-1+j}, DV_{p-n-1+k}\right)$$

$$= \frac{dt}{(2n+1)^2} \sum_{j=1}^{2n+1} \sum_{k=1}^{2n+1} \min(p-n-1+j, p-n-1+k)$$

$$= \frac{dt}{(2n+1)^2} \left[ (2n+1)^2(p-n-1) + \sum_{j=1}^{2n+1} \sum_{k=1}^{2n+1} \min(j,k) \right]$$

Given that $\sum_{j=1}^{n} \sum_{k=1}^{n} \min(j,k) = \sum_{j=1}^{n} j^2 = n(n+1)(2n+1)/6$

$$\text{Var}\left(MDV_p^n\right) = \frac{dt}{(2n+1)^2} \left[ (2n+1)^2(p-n-1) + (2n+1)(2n+2)(4n+3)/6 \right]$$

$$= \frac{3p + 6np - 2n - 2n^2}{3(2n+1)} dt$$

Therefore, the correlation

$$\text{Corr}\left(MDV_p^n, MDV_q^n\right) = \frac{\text{Cov}\left(MDV_p^n, MDV_q^n\right)}{\sqrt{\text{Var}\left(MDV_p^n\right)\text{Var}\left(MDV_q^n\right)}}$$

$$= \frac{3p(2n+1)}{\sqrt{\left(3p + 6np - 2n - 2n^2\right)\left(3q + 6nq - 2n - 2n^2\right)}}$$

In contrast for the point estimates at $p$ and $q$

$$\text{Corr}\left(DV_p, DV_q\right) = \sqrt{\frac{p}{q}}$$

It is useful to re-express the above two equations in terms of actual time $t_p$ and $t_q$ and window size $t_n$. Substituting for $p$, $q$, and $n$ with $t_p/dt$, $t_q/dt$ and $t_n/dt$

$$\text{Corr}\left(MDV_{t_p}^{t_n}, MDV_{t_q}^{t_n}\right) = \frac{3t_p\left(2t_n + dt\right)}{\sqrt{\left(3t_p dt + 6t_n t_p - 2t_n dt - 2t_n^2\right)\left(3t_q dt + 6t_n t_q - 2t_n dt - 2t_n^2\right)}}$$

$$= \frac{3t_p\left(2t_n + dt\right)}{\sqrt{\left(\left(3t_p - 2t_n\right)dt + 6t_n t_p - 2t_n^2\right)\left(\left(3t_q - 2t_n\right)dt + 6t_n t_q - 2t_n^2\right)}}$$

$$\text{Corr}\left(DV_{t_p}, DV_{t_q}\right) = \sqrt{\frac{t_p}{t_q}}$$

