## [Editor Report · eLife assessment]

This **fundamental** work quantifies the stochastic dynamics of neural population activity in the lateral intraparietal area (LIP) of the macaque monkey brain during single perceptual decisions. These single-trial dynamics have been subject to intense debate in neuroscience, and they have significant implications for modeling decision-making in various fields including neuroscience and psychology. Through a combination of state-of-the-art recordings from many LIP neurons and theory-driven data analyses, the authors provide **convincing** evidence for the notion that single-trial neural population dynamics in LIP encode the decision variable postulated by the drift-diffusion model of decision-making.

---

## [Referee Report · Reviewer #1 (Public Review)]

Summary:

In this paper, Steinemann et al. characterized the nature of stochastic signals underlying the trial-averaged responses observed in lateral intraparietal cortex (LIP) of non-human primates (NHPs), while these performed the widely used random dot direction discrimination task. Ramp-up dynamics in the trial averaged LIP responses were reported in numerous papers before. But the temporal dynamics of these signals at the single-trial level have been subject to debate. Using large scale neuronal recordings with Neuropixels in NHPs, allows the authors to settle this debate rather compellingly. They show that drift-diffusion like computations account well for the observed dynamics in LIP.

Strengths:

This work uses innovative technical approaches (Neuropixel recordings in behaving macaque monkeys). The authors tackle a vexing question that requires measurements of simultaneous neuronal population activity and hence leverage this advanced recording technique in a convincing way.

They use different population decoding strategies to help interpret the results.

They also compare how decoders relying on the data-driven approach using dimensionality reduction of the full neural population space compares to decoders relying on more traditional ways to categorize neurons that are based on hypotheses about their function. Intriguingly, although the functionally identified neurons are a modest fraction of the population, decoders that only rely on this fraction achieve comparable decoding performance to those relying on the full population. Moreover, decoding weights for the full population did not allow the authors to reliably identify the functionally identified subpopulation.

The revision addressed the minor weaknesses to our satisfaction.

---

## [Referee Report · Reviewer #2 (Public Review)]

Steinemann, Stine, and their co-authors studied the noisy accumulation of sensory evidence during perceptual decision-making using Neuropixels recordings in awake, behaving monkeys. Previous work has largely focused on describing the neural underpinnings through which sensory evidence accumulates to inform decisions, a process which on average resembles the systematic drift of a scalar decision variable toward an evidence threshold. The additional order of magnitude in recording throughput permitted by the methodology adopted in this work offers two opportunities to extend this understanding. First, larger-scale recordings allow for the study of relationships between the population activity state and behavior without averaging across trials. The authors' observation here of covariation between the trial-to-trial fluctuations of activity and behavior (choice, reaction time) constitutes interesting new evidence for the claim that neural populations in LIP encode the behaviorally-relevant internal decision variable. Second, using Neuropixels allows the authors to sample LIP neurons with more diverse response properties (e.g. spatial RF location, motion direction selectivity), making the important question of how decision-related computations are structured in LIP amenable to study. For these reasons, the dataset collected in this study is unique and potentially quite valuable. This revised manuscript addresses a number of questions regarding analyses which were unclear in the original manuscript, and as a result the study is a strong contribution toward our understanding of neural mechanisms of decision making.

---

## [Author Response]

The following is the authors’ response to the original reviews.

**Public Reviews:**

**Reviewer #1 (Public Review):**
Summary:In this paper, Steinemann et al. characterized the nature of stochastic signals underlying the trial-averaged responses observed in the lateral intraparietal cortex (LIP) of non-human primates (NHPs), while these performed the widely used random dot direction discrimination task. Ramp-up dynamics in the trial averaged LIP responses were reported in numerous papers before. However, the temporal dynamics of these signals at the single-trial level have been subject to debate. Using large-scale neuronal recordings with Neuropixels in NHPs, allows the authors to settle this debate rather compellingly. They show that drift-diffusion-like computations account well for the observed dynamics in LIP.Strengths:This work uses innovative technical approaches (Neuropixel recordings in behaving macaque monkeys). The authors tackle a vexing question that requires measurements of simultaneous neuronal population activity and hence leverage this advanced recording technique in a convincing wayThey use different population decoding strategies to help interpret the results.They also compare how decoders relying on the data-driven approach using dimensionality reduction of the full neural population space compare to decoders relying on more traditional ways to categorize neurons that are based on hypotheses about their function. Intriguingly, although the functionally identified neurons are a modest fraction of the population, decoders that only rely on this fraction achieve comparable decoding performance to those relying on the full population. Moreover, decoding weights for the full population did not allow the authors to reliably identify the functionally identified subpopulation.Weaknesses:No major weaknesses beyond a few, largely clarification issues, detailed below.

We thank Reviewer 1 (R1) for this summary. The revised manuscript incorporates R1’s suggestions, as detailed below.

**Reviewer #2 (Public Review):**
Steinemann, Stine, and their co-authors studied the noisy accumulation of sensory evidence during perceptual decision-making using Neuropixels recordings in awake, behaving monkeys. Previous work has largely focused on describing the neural underpinnings through which sensory evidence accumulates to inform decisions, a process which on average resembles the systematic drift of a scalar decision variable toward an evidence threshold. The additional order of magnitude in recording throughput permitted by the methodology adopted in this work offers two opportunities to extend this understanding. First, larger-scale recordings allow for the study of relationships between the population activity state and behavior without averaging across trials. The authors’ observation here of covariation between the trial-to-trial fluctuations of activity and behavior (choice, reaction time) constitutes interesting new evidence for the claim that neural populations in LIP encode the behaviorally-relevant internal decision variable. Second, using Neuropixels allows the authors to sample LIP neurons with more diverse response properties (e.g. spatial RF location, motion direction selectivity), making the important question of how decision-related computations are structured in LIP amenable to study. For these reasons, the dataset collected in this study is unique and potentially quite valuable.However, the analyses at present do not convincingly support two of the manuscript’s key claims: (1) that ‘sophisticated analyses of the full neuronal state space’ and ‘a simple average of Tconin neurons’ yield roughly equivalent representations of the decision variable; and (2) that direction-selective units in LIP provide the samples of instantaneous evidence that these Tconin neurons integrate. Supporting claim (1) would require results from sophisticated population analyses leveraging the full neuronal state space; however, the current analyses instead focus almost exclusively on 1D projections of the data. Supporting claim (2) convincingly would require larger samples of units overlapping the motion stimulus, as well as additional control analyses.

We thank the reviewer (R2) for their careful reading of our paper and the many useful suggestions.

As detailed below, the revised manuscript incorporates new control analyses, improved quantification, and statistical rigor, which now provide compelling support for key claim #1. We do not regard claim #2 as a key claim of the paper. It is an intriguing finding with solid support, worthy of dissemination and further investigation. We have clarified the writing on this matter.

Specific shortcomings are addressed in further detail below:(1) The key analysis-correlation between trial-by-trial activity fluctuations and behavior, presented in Figure 5 is opaque, and would be more convincing with negative controls. To strengthen the claim that the relationship between fluctuations in (a projection of) activity and fluctuations in behavior is significant/meaningful, some evidence should be brought that this relationship is specific - e.g. do all projections of activity give rise to this relationship (or not), or what level of leverage is achieved with respect to choice/RT when the trial-by-trial correspondence with activity is broken by shuffling.

We do not understand why R2 finds the analysis opaque, but we are grateful for the lucid recommendations. The relationships between fluctuations in neural activity and behavior are indeed ‘specific’ in the sense that R2 uses this term. In addition to the shuffle control, which destroys both relationships (Reviewer Figure 1), we performed additional control analyses that preserve the correspondence of neural signals and behavior on the same trial. We generated random coding directions (CDs) by establishing weight vectors that were either chosen from a standard normal distribution or by permuting the weights assigned to PC-1 in each session. The latter is the more conservative measure. Projections of the neural responses onto these random coding directions render 𝑆rand(𝑡). Specifically, the degree of leverage is effectively zero or greatly reduced. These analyses are summarized in a new Supplementary Figure S10. The bottom row of Figure S10 also addresses the question, “What degree of leverage and mediation would be expected for a theoretical decision variable?” This is accomplished by simulating decision variables using the drift-diffusion model fits in Figure 1c. The simulation is consistent with the leverage and (incomplete) mediation observed for the populations of Tcon neurons. For details see Methods, Simulated decision variables and Leverage of single-trial activity on behavior.

(2) The choice to perform most analysis on 1D projections of population activity is not wholly appropriate for this unique type of dataset, limiting the novelty of the findings, and the interpretation of similarity between results across choices of projection appears circular:

We disagree with the characterization of our argument as circular, but R2 raises several important points that will probably occur to other careful readers. We address them as subpoints 2.1–2.4, below. Importantly, we are neither claiming nor assuming that the LIP population activity is one-dimensional. We have revised the paper to avoid giving this impression. We are also not claiming that the average of Tin neurons (or the 1D projections) explains all features of the LIP population, nor would we expect it to, given the diversity of response fields across the population. Our objective is to identify the specific dimension within population activity that captures the decision variable (DV), which has been characterized successfully as a one-dimensional stochastic process—that is, a scalar function of time. We have endeavored to clarify our thinking on this point in the revised manuscript (e.g., lines 97–98, 103–104).

(2.1) The bulk of the analyses (Figure 2, Figure 3, part of Figure 4, Figure 5, Figure 6) operate on one of several 1D projections of simultaneously recorded activity. Unless the embedding dimension of these datasets really does not exceed 1 (dimensionality using e.g. participation ratio in each session is not quantified), it is likely that these projections elide meaningful features of LIP population activity.

We now report the participation ratio (4.4 ± 0.4, mean ± s.e. across sessions), and we state that the first 3 PCs explain 67.1±3.1% of the variance of time- and coherence-dependent signals used for the PCA. We agree that the 1D projections may elide meaningful features of LIP population activity. Indeed, we make this point through our analysis of the Min neurons. We do not claim that the 1D projections explain all of the meaningful features of LIP population activity. They do, however, reveal the decision variable, which is our main focus. These 1D signals contain features that correlate with events in the superior colliculus, summarized in Stine et al. (2023), attesting to their biological relevance.

(2.2) Further, the observed similarity of results across these 1D projections may not be meaningful/interpretable. First, the rationale behind deriving Sramp was based on the ramping historically observed in Tin neurons during this task, so should be expected to resemble Tin.

The Reviewer is correct that we would expect 𝑆ramp to resemble the ramping observed in Tin neurons. We refer to this approach as hypothesis-driven. It captures the drift component of drift-diffusion. It is true that the Tcon neurons exhibit such ramps in their trial average firing rates, but this does not guarantee in that the single-trial population firing rates would manifest as drift-diffusion. Indeed Latimer et al. (2015) concluded that the ramp-like averages comprise stepping from a low to a high firing rate on each trial at a random time. Therefore, while R2 is right to characterize the similarity of Tcon to the ramp direction in in trial-averaged activity as unsurprising, their similarity on single trials is not guaranteed.

(2.3) Second, Tin comprises the largest fraction of the neuron groups sampled during most sessions, so SPC1 should resemble Tin too. The finding that decision variables derived from the whole population’s activity reduce essentially to the average of Tin neurons is thus at least in part ’baked in’ to the approach used for deriving the decision variables.

This is incorrect. The Tcon in neurons constitute only 14.5% of the population, on average, across the sessions (see Table 1). This misunderstanding might contribute to R2’s concern about the importance of these neurons in shaping PC1. It is not simply because they are over-represented. Also, addressing R2’s concern about circularity, we would like to remind R2 that the selection of Tin neurons was based only on their spatial selectivity in the delayed saccade task. We do not see how it could be baked-in/guaranteed that a simple average of these neurons (i.e. zero degrees of freedom) yields dynamics and behavioral correlations that match those produced by dimensionality-reduction techniques that (𝑖) have degrees of freedom equal to the number of neurons and (𝑖𝑖) are blind to the neurons’ spatial selectivity. We have additionally modified what is now Supplementary Figure S13 (old Supplementary Figure S8), which portrays the mean accuracy of choice decoders trained on the neural activity of all neurons, only Tin neurons, all but the Tin neurons, and all but Tin and Min neurons, respectively. Figure S13 now highlights how much more readily choice can be decoded from the small population of Tin neurons than the remainder of the population.

(2.4) The analysis presented in Figure S6 looks like an attempt to demonstrate that this isn’t the case, but is opaque. Are the magnitudes of weights assigned to units in Tin larger than in the other groups of units with preselected response properties? What is their mean weighting magnitude, in comparison with the mean weight magnitude assigned to other groups? What is the null level of correspondence observed between weight magnitude and assignment to Tin (e.g. a negative control, where the identities of units are scrambled)?

The revised Figure S6—what is now Figure S9—displays more clearly that the weights assigned to Tcon and Tips neurons (purple & yellow, respectively) are larger in magnitude than those assigned in in to other neurons (gray). Author response table 1 shows a more detailed breakdown of the groups. Note that the length of the vector of weights is one. We are unsure what R2 means by ‘the null level of correspondence.’ Perhaps it helps to know that the mean weight of the ‘other neurons’ is close to zero for all four coding directions. However, it is the overlap of the weights and the relative abundance of non-Tin neurons that is more germane to the point we are making. To wit, knowing the weight (or percentile) of a neuron is a poor predictor that it belongs to the Tin category. This point is most clearly supported by the logistic regression (Fig. S9, bottom row). In other words, the large group of non-Tin neurons contribute substantially to all four coding directions examined in Figure S9. Thus, the similarity between Tin neurons and PC1 is not simply due to an over-representation of Tin neurons as suggested in item 2.3.

**Author response table 1. sa3table1:** Mean weights assigned to neuron classes in four coding directions.

Neuron class	Sramp	SPC1	SWhen	SWhat
wTinC	0.073	0.093	0.063	0.059
wTinI	-0.035	-0.063	-0.027	-0.022
wMinL	0.011	0.012	0.028	0.027
wMinR	-0.035	-0.043	-0.027	0.014
wother	0.0010	0.0051	0.0020	0.0070

(3) The principal components analysis normalization procedure is unclear, and potentially incorrect and misleading: Why use the chosen normalization window (±25ms around 100ms after motion stimulus onset) for standardizing activity for PCA, rather than the typical choice of mean/standard deviation of activity in the full data window? This choice would specifically squash responses for units with a strong visual response, which distorts the covariance matrix, and thus the principal components that result. This kind of departure from the standard procedure should be clearly justified: what do the principal components look like when a standard procedure is used, and why was this insufficient/incorrect/unsuitable for this setting?

We used the early window because it is a robust measure of overall excitability, but we now use a more conventional window that spans the main epoch of our analyses, 200–600 ms after motion onset. This method yields results qualitatively similar to the original method. We are persuaded that this is the more sensible choice. We thank R2 for raising this concern.

(4) Analysis conclusions would generally be stronger with estimates of variability and control analyses: This applies broadly to Figures 2-6.

We have added estimates of variability and control analyses where appropriate.

Figure 2 shows examples of single-trial signals. The variability is addressed in Figure 3a and the new Supplementary Figure S5.

Figure 3 now contains error bars derived by bootstrapping (see Methods, Variance and autocorrelation of smoothed diffusion signals). We have also added Supplementary Figure S5, which substantiates the sublinearity claim using simulations.

Figure 4 (i) We now indicate the s.e.m. of decoding accuracy (across sessions) by the shading in Figure 4a. (ii) The black symbols in new Supplementary Figure S8 show the mean ± s.e.m. for all pairwise comparisons shown in Figure 4d & e. (iii) Supplementary Figure S8 also summarizes two control analyses that deploy random coding directions (CDs) in neuronal state space. The upper row of Fig S9 compares the observed cosine similarity (CoSim)—between the CD identified by the graph title and the other four CDs labeled along the abscissa—with values obtained with 1000 random CDs established by random permutations of the weight assignments. The brown symbols are the mean ± sdev of the CoSim (N=1000). The error bars are smaller than the symbols. We use the cumulative distribution of CoSim under permutation to estimate p-values (p<0.001 for all comparisons). We used a similar approach to estimate the distribution of the analogous correlation statistics between signals rendered by random directions in state space (Figure S8, lower row). For additional details, please see Methods, Similarity of single-trial signals.

Figure 5: The rigor of all claims associated with this figure is adduced from two control analyses and a simulation. The first control breaks the trial-by-trial correspondence between neural signals and behavior (Reviewer Figure 1). The second control shows that neural activity does not have substantial leverage on behavior when projected onto random directions in state space (Supplementary Figure S10, top). Simulations of decision variables using parameters derived from the fits to the behavioral data (Figure 1) support a degree of leverage and mediation comparable to the values observed for 𝑆Tincon (Supplementary Figure S10, bottom). For additional details, please see Methods (Leverage of single-trial activity on behavior) and the reply to item 1, above.

Figure 6: Panels c&d show estimates of variability across neurons and experimental sessions, respectively. The reported p-value is based on a permutation test (see Methods, Correlations between Min and Tconin). The correlations shown in panel e (heatmap) are derived from pooled data across sessions. The reported p-value is based on a permutation test (see Methods, Correlations between Min and Tconin).

**Reviewer #3 (Public Review):**
Summary:The paper investigates which aspects of neural activity in LIP of the macaque give rise to individual decisions (specificity of choice and reaction times) in single trials, by recording simultaneously from hundreds of neurons. Using a variety of dimensionality reduction and decoding techniques, they demonstrate that a population-based drift-diffusion signal, which relies on a small subset of neurons that overlap choice targets, is responsible for the choice and reaction time variability. Analysis of direction-selective neurons in LIP and their correlation with decision-related neurons (T con in [Tconin] neurons) suggests that evidence integration occurs within area LIP.Strengths:This is an important and interesting paper, which resolves conflicting hypotheses regarding the mechanisms that underlie decision-making in single trials. This is made possible by exploiting novel technology (Primatepixels recordings), in conjunction with state-of-the-art analyses and well-established dynamic random dot motion discrimination tasks.General recommendations(1) Please tone down causal language. You present compelling correlative evidence for the idea that LIP population activity encodes the drift-diffusion DV. We feel that claims beyond that (e.g., ‘Single-trial drift-diffusion signals control the choice and decision time’) would require direct interventions, and are only partially supported by the current evidence. Further examples are provided in point 1 of Reviewer 1 below.

We have adopted the recommendation to ‘tone down the causal language.’ Throughout the manuscript, we strive to avoid conveying the false impression that the present findings provide causal support for the decision mechanism. However, other causal studies of LIP support causality in the random dot motion task (Hanks et al., 2006; Jeurissen et al., 2022). It is therefore justifiable to use terms that imply causality in statements intended to convey hypotheses about mechanism. We agree that we should not give the false impression that the present support for said mechanism is adduced from causal perturbations in this study, as there were none.

(2) Please provide a commonly used, data-driven quantification of the dimensionality of the population activity – for example, using participation ratio or the number of PCs explaining 90% of the variance. This will help readers evaluate the conclusions about the dimensionality of the data.

Principal component analysis reveals a participation ratio of 4.4 ± 0.4 (mean ±s.e., across sessions), and the first 3 PCs explain 67.1 ± 3.1 percent of the variance. The dimensionality of the data is low, but greater than one. We state this in Methods (Principal Component Analysis) and in Results (Single-trial drift-diffusion signals approximate the decision variable, lines 200–201).

(3) Please justify the normalization procedure used for PCA: Why use the chosen normalization window (±25ms around 100ms after motion stimulus onset) for standardizing activity for PCA, rather than the more common quantification of mean/standard deviation across the full data window? What do the first principal components look like when the latter procedure is used?

We now use a more conventional window that spans the main epoch of our analyses, 200–600 ms after motion onset. This method yields results qualitatively similar to the original method. We are persuaded that this is the more sensible choice.

(4) Please provide estimates of variability for variance and autocorrelation in Fig. 3 (e.g., through bootstrapping). Further, simulations could substantiate the claim about the expected sub-linearity at later time points (Fig. 3a) due to the upper stopping bound and limited firing rate range.

We thank the reviewers for these helpful recommendations. The revised Fig. 3 now contains error bars derived by bootstrapping (see Methods, Variance and autocorrelation of smoothed diffusion signals). We have also added Supplementary Figure S5, which substantiates the sub-linearity claim using simulations.

(5) Please add controls and estimates of variability for decoding across sessions in Fig. 4: what are the levels of within-trial correlation/cosine similarity for random coding directions? What is the variability in the estimates of values shown in a/d/e?

We have addressed each of these items. (1) Figure 4a now shows the s.e.m. of decoding accuracy (across sessions). (2) Regarding the variability of estimates shown in Figure 4d & e, the standard errors are displayed in the new supplementary Figure S8. It makes sense to show them there because there is no natural way to represent error on the heat maps in Figure 4, and Figure S8 concerns the comparison of the values in Figure 4d&e to values derived from random coding directions. (3) Random coding directions lead to values of cosine similarity and within-trial correlation that do not differ significantly from zero. We show this in several ways, summarized in our reply to Public Review item 4. Additional details are in the revised manuscript (Methods, Similarity of single-trial signals) and the new Supplementary Figure S8.

(6) Please perform additional analysis to strengthen the claim from Fig. 6, that Min represents the integrand and not the integral. The analysis in Fig. 6d could be repeated with the integral (cumulative sum) of the single-trial Min signals. Does this yield an increase in leverage over time?

The short answer is, yes in part. Reviewer Figure 2a provides support for leverage of the integral on choice, and this leverage, like 𝑆Tincon (t), increases as a function of time. The effect is present in all seven sessions that have both Mleftin and Mrightin neurons (all 𝑝 < 1𝑒 − 10). However, as shown in panel b, the same integral fails to demonstrate more than a hint of leverage on RT. All correlations are barely negative, and the magnitude does not increase as a function of time. We suspect—but cannot prove—that this failure arises because of limited power and the expected weak effect. Recall that the mediation analysis of RT is restricted to longer trials. Moreover, the correlation between the Min difference and the Tin signal is less than 0.1 (heatmap, Fig. 6e), implying that the Min difference explains less than 1% of the variance of 𝑆Tin(𝑡). We considered including Reviewer Figure 2 in the paper, but we feel it would be disingenuous (cherry-picking) to report only the positive outcome of the leverage on choice. If the editors feel strongly about it, we would be open to including it, but leaving these analyses out of the revised manuscript seems more consistent with our effort to deëmphasize this finding. In the future, we plan to record simultaneously from populations MT and LIP neurons (Min and Tin, of course) and optimize Min neuron yield by placing the RDM stimulus in the periphery.

(7) Please describe the complete procedure for determining spatially-selective activity. E.g.: What response epoch was used, what was the spatial layout of the response targets, were responses to all ipsi- vs contralateral targets pooled, what was the spatial distribution of response fields relative to the choice targets across the population?

We thank the reviewers for pointing out this oversight. We now explain this procedure in the Methods (lines 629–644):

Neurons were classified post hoc as Tin by visual-inspection of spatial heatmaps of neural activity acquired in the delayed saccade task. We inspected activity in the visual, delay, and perisaccadic epochs of the task. The distribution of target locations was guided by the spatial selectivity of simultaneously recorded neurons in the superior colliculus (see Stine 2023 for details). Briefly, after identifying the location of the SC response fields, we randomly presented saccade targets within this location and seven other, equally spaced locations at the same eccentricity. In monkey J we also included 1–3 additional eccentricities, spanning 5–16 degrees. Neurons were classified as Tin if they displayed a clear, spatially-selective response in at least one epoch to one of the two locations occupied by the choice targets in the main task. Neurons that switched their spatial selectivity in different epochs were not classified as Tin. The classification was conducted before the analyses of activity in the motion discrimination task. The procedure was meant to mimic those used in earlier single-neuron studies of LIP (e.g., Roitman & Shadlen 2002) in which the location of the choice targets was determined online by the qualitative spatial selectivity of the neuron under study. The Tcon neurons in the in present study were highly selective for either the contralateral or ipislateral choice target used in the RDM task (AUC = 0.89±0.01; 𝑝 < 0.05 for 97% of neurons, Wilcoxon rank sum test). Given the sparse sampling of saccade target locations, we are unable to supply a quantitative estimate of the center and spatial extent of the RFs.

(8) Please clarify if a neuron could be classified as both Tin and Min. Or were these categories mutually exclusive?

These categories are mutually exclusive. If a neuron has spatially-selective persistent activity, as defined by the method described above, it is classified as a Tin neuron and not as an Min neuron even if it also shows motion-selective activity during passive motion viewing. We now specify this in the Methods (lines 831–832).

**Reviewer #1 (Recommendations for the Authors):**
𝑅∗1.1a Causal language (Line 23-24): ‘population activity represents […] drift’ and “we provide direct support for the hypothesis that drift-diffusion signal is the quantity responsible for the variability in choice and RT” reads at first sight as if the authors claim that they present evidence for a causal effect of LIP activity on choice. The authors are other wise nuanced and careful to point out that their evidence is correlational. What seems to be meant is that the population activity/drift-diffusion signal ‘approximates the DV that gives rise to the choices […]’ (cf. line 399). I would recommend using such alternative phrasing to avoid confusion (and the typically strong reactions by readers against misleading causal statements).

We have adopted the reviewer’s recommendation and have modified the text throughout to reduce causal language. See our response to General Recommendation 1.

𝑅∗1.1b Relatedly, any discussion about the possibility of LIP being causally involved in evidence integration (e.g. lines 429-445 [Au: now 462–478]) should also comment on the possibility of a distributed representation of the decision variable given that neural correlates of the DV have been reported in several areas including PFC, caudate and FEF.

We believe this is possible. However, we hope to avoid discussions about causality given that it is not a focus of the paper. Although it is somewhat tangential, we have shown elsewhere that LIP is causal in the sense that causal manipulations affect behavior, but it is also true that causality does not imply necessity, and similarly, lack of necessity does not imply ‘only correlation.’ Regarding distributed representations, it is worth keeping in mind the cautionary counter-example furnished by the SC study (Stine et al., 2023). The firing rates measured by averaging over trials are similar in SC and LIP; both manifest as coherence and direction-dependent ramps, leading to the suggestion that they form a distributed representation of the decision variable. With single-trial resolution, we now know that LIP and SC exhibit distinct dynamics—drift-diffusion and bursting, respectively. It remains to be seen if single-trial resolution achievable by simultaneous Neuropixels recordings from prefrontal areas and LIP reveal shared or distinct dynamics.

𝑅∗1.2 How was the spatially selective activity determined? The classification of Tin neurons is critical to this study - how was their spatial selectivity determined? Please describe this in similar detail as the description of direction selectivity on lines 681-690 [Au: now 824–832]. E.g.: what response epoch was used, what was the spatial layout of the response targets, were responses to all ipsi- vs contralateral targets pooled, and what was the spatial distribution of response fields relative to the choice targets across the population?

We now explain the selection procedure in Methods (lines 629–644). Please see our reply to General Recommendation 7, above.

𝑅∗1.3 Could a neuron be classified as both Tin and Min, or were these categories mutually exclusive? Please clarify. (This goes beyond the scope of the current study: but did the authors find evidence for topographic organization or clustering of these categories of neurons?)

These categories are mutually exclusive. Please see our response to General Recommendation 8, above.

𝑅∗1.4 Contrary to the statement on line 121, the trial averages in Fig. 2a, 2b show coherence dependency at the time of the saccade in saccade-aligned traces for the coding strategies, except for STin (fig. 2c). Is this a result of the choice for t1 (= 0.1s)? (The authors may want to change their statement on line 121.) Relatedly, do the population responses for the two coding strategies Sramp and SPC1 depend on the epoch used to derive weights for individual neurons?

We have revised the description to accommodate R2’s observation. 𝑆ramp retains weak coherence-dependence before saccades towards the choice target contralateral to the recording site. This was true in four of the eight sessions. For 𝑆PC1, there is no longer a coherence dependency for the Tin choices, owing to the change in normalization method (see revised Figure 2b).

We also corrected an error in the Methods section. Specifically, the ramp ends at 𝑡1 = 0.05 s before the time of the saccade, not 𝑡1 = 0.1 s. While we no longer emphasize the similarity of traces aligned to saccade, it is reasonable to find issue with the observation that they retain a dependency on coherence (𝑆ramp only) because, according to theory, traces associated with Tin choices should reach a common positive threshold at decision termination. That said, for the Ramp direction there may be a reason to expect this discrepancy from theory. The deterministic part of drift-diffusion includes an urgency signal that confers positive convexity to the deterministic drift. This accelerating nonlinearity is not captured by the ramp, and it is more prominent at longer decision times, thus low coherences. We do not share this interpretation in the revised manuscript, in part because retention of coherence dependency is present in only half the sessions (see Reviewer Figure 3) The correction to the definition of 𝑡1 also provides an opportunity to address R2’s final question (‘Relatedly,…?’). For 𝑆ramp this particular variation in 𝑡1 does not affect 𝑆ramp, and 𝑆PC1 no longer retains coherence dependency for Tin choices. Note that our choice of 𝑡0 and 𝑡1 is based on the empirical observation that the ramping activity in response averages of Tin neurons typically begins 200 ms after motion onset and ends 50–100 ms before initiation of the saccadic choice. The starting time (𝑡0) is also supported by the observation that the decoding accuracy of a choice-decoder begins to diverge from chance at this time (Figure 4a).

𝑅∗1.5 It is intriguing that Sramp and SPC1 show dynamics that look so similar (fig. 2a, 2b). How do the weights assigned to each neuron in both strategies compare across the population?

The weights assigned to each neuron are very similar across the two strategies as indicated by a cosine similarity (0.65 ± 0.04, mean ± s.e.m. across sessions).

𝑅∗1.6 Tin neurons, which show dynamics closely resembling different coding directions (fig. 2) and the decoders do not have weights that can distinguish them from the rest of the population in each of these analyses (fig. S7). Is it fair to interpret these findings as evidence for broad decision-related co-variability in the recorded neural population in LIP?

Yes, our results are consistent with this interpretation. However, it is worth reiterating that decoding performance drops considerably when Tin neurons are not included (see Supplementary Figure S13). Thus, this broad decision-related co-variability is present but weak.

𝑅∗1.7 It is intriguing that the decoding weights of the different decoders did not allow the authors to reliably identify Tin neurons. Could this be, in part, due to the low dimensionality of the population activity and task that the animals are presumably overtrained on? Or do the authors expect this finding to hold up if the population activity and task were higher dimensional?

Great question! We can only speculate, but it seems possible that a more complex, ‘higher dimensional’ task could make it easier to identify Tin neurons. For example, a task with four choices instead of two may decrease correlations among groups of neurons with different response fields. We have added this caveat to the discussion (lines 459-–461). One minor semantic objection: The animal has learned to perform a highly contrived task at low signal-to-noise. The animal is well-trained, not over-trained.

𝑅∗1.8 Lines 135-137 [Au: now 141–142]: The similarity in the single trial traces from different coding strategies (fig. 2a-2c, left) is not as evident to me as the authors suggest. It might be worthwhile computing the correlation coefficients between individual traces for each pair of strategies and reporting the mean correlation to support the author’s point.

We report the mean correlation between single-trial signals generated by the chosen dimensionality reduction methods in Figure 4e. We show the variability in this measure in Supplementary Figure S8. We have also adjusted the opacity of the single-trial traces in Figure 2, left.

𝑅∗1.9 Minor/typos:-line 74: consider additionally citing Hyafil et al. 2023.-line 588: ‘that were strongly correlated’?-line 615: ‘were the actual drift-diffusion process were...’.-line 717: ‘a causal influence’ -> ‘no causal influence’.Fig. 6: panel labels e vs d are swapped between the figure and caption.Fig. 3c: labels r1,3 & r2,3 are flipped.

We have addressed all of these items. Thank you.

**Reviewer #2 (Recommendations for the Authors):**
𝑅∗2.1 (Figure 2) Determine whether restricting the analysis to 1D projections of the data is a suitable approach given the actual dimensionality of the datasets being analyzed:- Should show some quantification of the dimensionality of the recorded activity; could do this by quantifying the dimensionality of population activity in each session, e.g. with participation ratio or related measures (like # PCs to explain some high proportion of the variance, e.g. 90 %). If much of the variation is not described in 1 dimension, then the paper would benefit from some discussion/analysis of the signals that occupy the other dimensions.

We now report the participation ratio (4.4 ± 0.4, mean ±s.e. across sessions), and we state that the first 3 PCs explain 67.1 ± 3.1% of the variance of the time- and coherence-dependent signals used for the PCA (mean ±s.e). We agree that the 1D projections may elide meaningful features of LIP population activity. Indeed, we make this point through our analysis of the Min neurons. To reiterate our response above, we do not claim that the 1D projections explain all of the meaningful features of LIP population activity. They do, however, reveal the decision variable, which is our main focus. These 1D signals contain features that correlate with events in the superior colliculus, summarized in Stine et al. (2023), attesting to their biological relevance.

The Reviewer is correct that our approach presupposes a linear embedding of the 1D decision variable in the population activity. In other words, an onlinear representation of the 1D decision variable in population activity could have an embedding dimensionality greater than 1, and there may well be a non-linear method that reveals this representation. To test this possibility, we decoded choice on each trial from population activity using (1) a linear decoder (logistic classifier) or (2) a multi-layer neural network, which can exploit non-linearities. We found that, for each session, the two decoders performed similarly: the neural network outperforms the logistic decoder (barely) in just one session. The analysis suggests that the assumption of linear embedding of the decision variable is justified. We hope this analysis convinces the reviewer that ‘sophisticated analyses of the full neuronal state space’ and ‘a simple average of [Tcon] neurons’ do in indeed yield roughly equivalent representations of the decision variable. We have included the results of this analysis in Supplementary Figure S12. See also item 2 of the Public response.

𝑅∗2.2 (Figure 3) Add estimates of variability for variance and autocorrelation through time from single-trial signals:– E.g. by bootstrapping. Would be helpful for making rigorous the discussion of when the deviation from the theory is outside what would be expected by chance, even if it doesn’t change the specific conclusions here.– If possible, it would help (by simulations, or maybe an added reference if it exists) to substantiate the claim about the expected sub-linearity at later time-points (Figure 3a) due to the upper stopping bound and limited firing rate range.

We thank the reviewer for this helpful comment. The revised Fig. 3 now contains error bars derived by bootstrapping (see Methods, Variance and autocorrelation of smoothed diffusion signals). We have also added Supplementary Figure S5, which substantiates the sub-linearity claim using simulations.

𝑅∗2.3 (Figure 4) Add controls and estimates of variability for decoding across sessions:– As a baseline - what is the level of within-trial correlation/cosine similarity when random coding directions are used?– What is the variability in the estimates of values shown in a/d/e?

We have addressed each of these items. (1) Figure 4a now shows the s.e.m. of decoding accuracy (across sessions). (2) Regarding the variability of estimates shown in Figure 4d & e, the standard errors are displayed in the new Supplementary Figure S8. It makes sense to show them there because (i) there is no natural way to represent error on the heat maps in Figure 4, and (ii) S8 concerns the comparison of the values in Figure 4d & e to values derived from random coding directions. (3) Random coding directions lead to values of cosine similarity and within-trial correlation that do not differ significantly from zero. We show this in several ways, summarized in our reply to Public Review item 4. Additional details are in the revised manuscript (Methods: Similarity of single-trial signals) and the new Supplementary Figure S8. We also provide this information in response to Recommendation 5, above.

𝑅∗2.4 (Figure 5) Add negative controls and significance tests to support claims about trends in leverage:– What is the level of increase in leverage attained from random 1D projections of the data, or other projections where the prior would be no leverage?– What is the range of leverage values fit for a simulated signal with a ground-truth of no trend?

We have added two control analyses. In addition to a shuffle control, which destroys the relationship (Review Figure 1) we performed additional analyses that preserve the correspondence of neural signals and behavior on the same trial. We generated random coding directions (CDs) by establishing weight-vectors that were either chosen from a Normal distribution or by permuting the weights assigned to PC-1 in each session. The latter is the more conservative measure. Projections of the neural responses onto these random coding directions render 𝑆rand(𝑡). Specifically, the degree of leverage is effectively zero or very much reduced. These analyses are summarized in a new Supplementary Figure S10. The distributions of our test statistics (e.g., leverage on choice and RT) under the variants of the null hypothesis also support traditional metrics of statistical significance. Figure S10 (bottom row) also provides an approximate answer to the question: What degree of leverage and mediation would be expected for a theoretical decision variable? Briefly, we simulated 60,000 trials using the race model that best fits the behavioral data of monkey M. For any noise-free representation of a Markovian integration process, the leverage of an early sample of the DV on behavior would be mediated completely by later activity as the latter sample—up to the time of commitment—subsumes all variability captured by the earlier sample. We, therefore, generated 𝑆sim(𝑡) by first subsampling the simulated data to match the trial numbers of each session. To evaluate a DV approximated from the activity of 𝑁 Tconin neurons per session rather than the true DV represented by the entire population, we generated 𝑁 noisy instantiations of the signal for each of the subsampled, simulated trials. The noisy decision variable, 𝑆sim (t) is the mean activity of these 𝑁 noise-corrupted signals. The simulation is consistent with the leverage and incomplete mediation observed for the populations of Tcon neurons. For in additional details, see Methods, Leverage of single-trial activity on behavior and Supplementary Figure S10, caption. See also our response to item 1 of the Public Response.

𝑅∗2.5 The analysis is performed across several signed coherence levels, with data detrended for each signed coherence and choice to enable comparison of fluctuations relative to the relevant baseline; are results similar for the different coherences?

The results are qualitatively similar for individual coherences. There is less power, of course, because there are fewer trials. The analyses cannot be performed for coherences ≥ 12.8% because there are not enough trials that satisfy the inclusion criteria (presence of left and right choice trials with RT ≤ 670 ms). Nonetheless, leverage on choice and RT is statistically significant for 27 of the 30 combinations of motion strengths < 12.8% × three signals (𝑆ramp, 𝑆PC1 and 𝑆Tin) × behavioral measures (RT and choice) (RT: all 𝑝 < 0.008, Fisher-z; choice: all 𝑝 < 0.05, t-test). The three exceptions are trials with 6.4% coherence rightward motion, which do not correlate significantly with RT on leftward choice trials. Reviewer Figure 4 shows the results of the leverage and mediation analyses, using only the 0% coherence trials.

𝑅∗2.6 (Figure 6) Additional analysis to strengthen the claim that Min represents the integrand and not the integral:a. Repeating the analysis in Figure 6d with the integral (cumulative sum) of the single-trial Min signals and instead observing a significant increase in leverage over time would be strong evidence for this interpretation. If you again see no increase, then it suggests that the activity of these units (while direction selective) may not be strongly yoked to behavior. This scenario (no increasing leverage of the integral of Min on behavior through time) also raises an intriguing alternative possibility: that the noise driving the ’diffusion’ of drift-diffusion here may originate in the integrating circuit, rather than just reflecting the complete integration of noise in the stream of evidence itself.b. Repeating the analysis in Figure 6d with the projection of the M subspace onto its own first PC e.g. take the union of units {Mrightin, Mleftin} [our {Minright ,Minleft }], do PCA just on those units’ single trial activities, identify the first PC, and project those activities on that dimension to obtain SPC1-M.c. Ameliorating the sample-size limitation by relaxing the criteria for inclusion in Min - performing the same analyses shown, but including all units with visual RFs overlapping the motion stimulus, irrespective of their direction selectivity.

a. Reviewer Figure 2a provides support for leverage of the integral on choice, and this leverage, like STin con(t), increases as a function of time. The effect is present in all seven sessions that have both Minleft  and Minright  neurons (all 𝑝 < 1𝑒 − 10). However, as shown in panel b, the same integral fails to demonstrate more than a hint of leverage on RT (all correlations are negative) and the magnitude does not vary as a function of time. We suspect—but cannot prove—that this failure arises because of limited power and the expected weak effect. Recall that the mediation analysis of RT is restricted to longer trials and that the correlation between the Min difference and the signal is less than 0.1 over the heatmap in Fig. 6e, implying that the Min difference explains less than 1% of the variance of 𝑆Tin(𝑡). We considered including Reviewer Figure 2 in the paper, but we feel it would be disingenuous (cherrypicking) to report only the positive outcome of the leverage on choice. If the editors feel strongly about it, we would be open to including it, but leaving these analyses out of the revised manuscript seems more consistent with our effort to deëmphasize this finding. In the future, we plan to record simultaneously from populations MT and LIP neurons (Min and Tin, of course) and optimize Min neuron yield by placing the RDM stimulus in the periphery. We also provide this information in response to Recommendation (6) above.

b. We tried the R’s suggestion to apply PCA to the union of Min neurons Minright , Minleft  , fully expecting PC1 to comprise weights of opposite sign for the right and left preferring neurons, but that is not what we observed. Instead, the direction selectivity is distributed over at least two PCs. We think this is a reflection of the prominence of other signals, such as the strong visual response and normalization signals (see Shushruth et al., 2018). In the spirit of the R’s suggestion, we also established an ‘evidence coding direction’ using a regression strategy similar to the Ramp CD applied to the union of Min neurons. The strategy produced a coding direction with opposite signed weights dominating the right and left subsets. The projection of the neural data on this evidence CD yields a signal similar to the difference variable used in Fig. 6e (i.e., signals that are approximately constant firing rates vs time and scale as a function of signed coherence). These unintegrated signals exhibit weak leverage on choice and RT, consistent with Figure 6d. However, the integrated signal has leverage on choice but not RT, similar to the integral of the difference signal in Reviewer Figure 2.

c. We do not understand the motivation for this analysis. We could apply PCA or dPCA (or the regression approach, described above) to the population of units with RFs that overlap the motion stimulus, but it is hard to see how this would test the hypothesis that direction-selective neurons similar to those in area MT supply the momentary evidence. As mentioned, we have very few Min neurons (as few as two in session 3). Future experiments that place the motion stimulus in the periphery would likely increase the yield of Min neurons and would be better suited to study this question. As such, we do not see the integrand-like responses of Min neurons as a major claim of the paper. Instead, we view it as an intriguing observation that deserves follow-up in future experiments, including simultaneous recordings from populations of MT and LIP neurons (Min and Tin, of course). We have softened the language considerably to make it clear that future work will be needed to make strong claims about the nature of Min neurons.

𝑅∗2.7 Other questions: Figure 2c is described as showing the average firing rate of units in Tconin on single trials, but must also incorporate some baseline subtraction (as the shown traces dip into negative firing rates). What base line is subtracted? Are these residual signals, as described for later figures, or is a different method used? (Presumably, a similar procedure is used also for Figure 2a/b, given that all single-trial traces begin at 0.). Is the baseline subtraction justified? If the dataset really does reflect the decision variable with single-trial resolution, eliminating the baseline subtraction when visualizing single-trial activity might actually help to make the point clearer: trials which (for any reason) begin with a higher projection on the particular direction that furnishes the DV would be predicted to reach the decision bound, at any fixed coherence, more quickly than trials with a smaller projection onto this direction.

We thank the reviewer for this comment. For each trial, the mean activity between 175 ms and 225 ms after motion onset was subtracted when generating the single-trial traces. The baseline subtraction was only applied for visualization to better portray the diffusion component in the signal. Unless otherwise indicated, all analyses are computed on non-baseline corrected data. We now describe in the caption of Figure 2 that ‘For visualization, single-trial traces were baseline corrected by subtracting the activity in a 50 ms window around 200 ms.’ Examples of the raw traces used for all follow-up analyses are displayed in Reviewer Figure 6.

**Reviewer #3 (Recommendations for the Authors):**
I only have a few comments to make the paper more accessible:𝑅∗3.1 I struggle to understand how the linear fitting from -1 to 1 was done. More detail about how the single cell single-trial activity was generated to possibly go from -1 to 1 or do I completely misunderstand the approach? I assume the data standardization does that job?

We have rephrased and added clarifying detail to the section describing the derivation of the ramp signal in the Methods (Ramp direction).

We applied linear regression to generate a signal that best approximates a linear ramp, on each trial, 𝑖, that terminates with a saccade to the choice-target contralateral to the hemisphere of the LIP recordings. The ramps are defined in the epoch spanning the decision time: each ramp begins at 𝑓𝑖(𝑡0) = −1, where 𝑡0 = 0.2 s after motion onset, and ends at 𝑓𝑖(𝑡1) = 1, where 𝑡1 = 𝑡sac − 0.05 s (i.e., 50 ms before saccade initiation). The ramps are sampled every 25 ms and concatenated using all eligible trials to construct a long saw-tooth function (see Supplementary Figure S2). The regression solves for the weights assigned to each neuron such that the weighted sum of the activity of all neurons best approximates the saw-tooth. We constructed a time series of standardized neural activity, sampled identically to the saw-tooth. The spike times from each neuron are represented as delta functions (rasters) and convolved with a non-causal 25 ms boxcar filter. The mean and standard deviation of all sampled values of activity were used to standardize the activity for each neuron (i.e., Z-transform). The coefficients derived by the regression establish the vector of weights that define 𝑆ramp. The algorithm ensures that the population signal 𝑆ramp(𝑡), but not necessarily individual neurons, have amplitudes ranging from approximately −1 to 1.

𝑅∗3.2 It is difficult to understand how the urgency signal is derived, to then generate fig S4.

The urgency signal is estimated by averaging 𝑆𝑥(𝑡) at each time point relative to motion onset, using only the 0% coherence trials. We have clarified this in the caption of Supplementary Figure S4.

**Author response image 1. sa3fig1:** Shuffle control for Fig.5. Breaking the within-trial correspondence between neural signal, 𝑆(𝑡), and choice suppresses leverage to near zero.

**Author response image 2. sa3fig2:** Leverage of the integrated difference signal Min left −Min right  on choice and RT. Traces are the average leverage across seven sessions. Same conventions as in Figure 5.

**Author response image 3. sa3fig3:** Trial-averaged 𝑆ramp activity during individual sessions. Same as Figure 2b for individual sessions for Monkey M (left) and Monkey J (right). The figure is intended to illustrate the consistency and heterogeneity of the averaged signals. For example, the saccade-aligned averages lose their association with motion strength before left (contra) choices in sessions 1, 2, 5, and 6 but retain the association in sessions 3, 4, 7, and 8.

**Author response image 4. sa3fig4:** Drift-diffusion signals have measurable leverage on choice and RT even when only 0%-coherence trials are included in the analysis.

**Author response image 5. sa3fig5:** Raw single-trial activity for three types of population averages. Representative single-trial activity during the first 300 ms of evidence accumulation using two motion strengths: 0% and 25.6% coherence toward the left (contralateral) choice target. Unlike in Figure 2 in the paper, single-trial traces are not baseline corrected by subtracting the activity in a 50 ms window around 200 ms. We highlight a number of trials with thick traces and these are the same trials in each of the rows.